# Amazon forests capture high levels of atmospheric mercury pollution from artisanal gold mining

Jacqueline R. Gerson [1,2 ✉], Natalie Szponar[3], Angelica Almeyda Zambrano[4], Bridget Bergquist[3], Eben Broadbent[4], Charles T. Driscoll [5], Gideon Erkenswick[6,7], David C. Evers[8], Luis E. Fernandez[9,10,11], Heileen Hsu-Kim [12], Giancarlo Inga[13], Kelsey N. Lansdale[14], Melissa J. Marchese[2,15], Ari Martinez[16], Caroline Moore[13], William K. Pan[2,15], Raúl Pérez Purizaca[17], Victor Sánchez[18], Miles Silman[9,10,11], Emily A. Ury [1], Claudia Vega[9,10,11], Mrinalini Watsa [7,13] & Emily S. Bernhardt[1]

Mercury emissions from artisanal and small-scale gold mining throughout the Global South exceed coal combustion as the largest global source of mercury. We examined mercury deposition and storage in an area of the Peruvian Amazon heavily impacted by artisanal gold mining. Intact forests in the Peruvian Amazon near gold mining receive extremely high inputs of mercury and experience elevated total mercury and methylmercury in the atmosphere, canopy foliage, and soils. Here we show for the first time that an intact forest canopy near artisanal gold mining intercepts large amounts of particulate and gaseous mercury, at a rate proportional with total leaf area. We document substantial mercury accumulation in soils, biomass, and resident songbirds in some of the Amazon's most protected and biodiverse areas, raising important questions about how mercury pollution may constrain modern and future conservation efforts in these tropical ecosystems.

[1] Department of Biology, Duke University, Durham, NC 27708, USA. [2] Duke Global Health Institute, Duke University, Durham, NC 27708, USA. [3] Department of Earth Sciences, University of Toronto, Toronto, ON M5S 3B1, Canada. [4] School of Forest Resources and Conservation, University of Florida, Gainesville, FL 32611, USA. [5] Department of Civil and Environmental Engineering, Syracuse University, Syracuse, NY 13244, USA. [6] Department of Molecular Microbiology, Washington University School of Medicine, St. Louis, MO 63110, USA. [7] Field Projects International, Escondido, CA 92029, USA. [8] Biodiversity Research Institute, Portland, ME 04103, USA. [9] Centro de Innovación Científica Amazónica (CINCIA), Puerto Maldonado, Peru. [10] Center for Energy, Environment, and Sustainability (CEES), Wake Forest University, Winston-Salem, NC 27109, USA. [11] Department of Biology, Wake Forest University, Winston-Salem, NC 27109, USA. [12] Department of Civil and Environmental Engineering, Duke University, Durham, NC 27708, USA. [13] San Diego Zoo Wildlife Alliance, San Diego, CA 92101, USA. [14] Environmental Science Program, Duke University, Durham, NC 27708, USA. [15] Nicholas School of the Environment, Duke University, Durham, NC 27708, USA. [16] Department of Biological Sciences, California State University, Long Beach, CA 90840, USA. [17] Universidad Nacional de Piura, Piura, Peru. [18] Instituto de Investigación en Ecología y Conservación (IIECOO), La Libertad, Peru. ✉email: jgerson1@gmail.com

A growing challenge to tropical forested ecosystems is artisanal and small-scale gold mining (ASGM). This form of gold mining occurs in over 70 countries, is frequently either informal or illegal, and accounts for approximately 20% of the world's gold production[1]. While ASGM is an important livelihood for local communities, it results in widespread deforestation[2,3], extensive conversion of forests to ponds[4], high sediment loading in nearby rivers[5,6], and is the largest global source of atmospheric mercury (Hg) emissions and freshwater Hg releases[7]. Many intensive ASGM sites are within global biodiversity hotspots and lead to decreased diversity[8], loss of sensitive species[9], and high exposure of Hg in both people[10–12] and top predators[13,14]. An estimated 675–1000 tons Hg yr$^{-1}$ are volatilized and emitted to the atmosphere globally from ASGM operations[7]. This use of enormous quantities of Hg in ASGM has shifted the major emission source of atmospheric Hg from the Global North to the Global South, with consequences for patterns in Hg fate, transport, and exposure. However, little is known about the fate of these atmospheric Hg emissions and patterns of deposition and accumulation across ASGM-impacted landscapes.

The international Minamata Convention on Mercury entered into force in 2017 with Article 7 specifically directed at Hg releases from ASGM. In ASGM, liquid elemental Hg is added to sediment or ores to isolate gold. This amalgam is subsequently heated, which concentrates the gold and releases gaseous elemental Hg (GEM; Hg$^0$) into the atmosphere. Amalgam burning often occurs without a retort or other Hg-capturing device, though efforts are being undertaken by groups such as the United Nations Environment Program (UNEP) Global Mercury Partnership, United Nations Industrial Development Organization (UNIDO), and non-governmental organizations to encourage miners to mitigate Hg emissions. As of this writing in 2021, 132 countries including Peru have signed onto the Minamata Convention and have begun to develop National Action Plans to specifically address Hg reductions in association with ASGM. Scholars have called on these National Action Plans to be inclusive, ongoing, and holistic, considering both socioeconomic drivers and environmental harms[15–18]. Current plans to address the consequences of Hg in the environment focus on Hg risks associated with ASGM near aquatic ecosystems, involving miners and those living near amalgam burning, and involving communities that consume large quantities of predatory fish. Occupational Hg exposure via inhalation of Hg vapors from amalgam burning, dietary Hg exposure via the consumption of fish, and bioaccumulation of Hg in the aquatic food web have been the focus of most scientific studies related to ASGM, including early studies in the Amazon (e.g., see Lodenius and Malm[19]).

Terrestrial ecosystems are also at risk of Hg exposure from ASGM. There are three main pathways by which atmospheric Hg released from ASGM as GEM can return to the terrestrial landscape[20] (Fig. 1): GEM can sorb to particles in the atmosphere, which are then intercepted by surfaces; GEM can be taken up directly by plants and incorporated into their tissues; finally, GEM can be oxidized to Hg(II) species, which can be dry deposited, sorbed to atmospheric particles, or entrained in rainwater. These pathways supply Hg to soils via throughfall (i.e., precipitation that passes through the canopy), litterfall, and rainfall, respectively. Wet deposition can be determined by Hg flux in precipitation collected in clearings. Dry deposition can be determined by the sum of Hg flux in litterfall and throughfall minus Hg flux in precipitation[21]. Numerous studies document Hg enrichment in terrestrial and aquatic ecosystems immediately adjacent to ASGM activity (e.g., see summary table in Gerson et al.[22]), which likely results from both depositional Hg inputs and direct Hg releases. However, while enhanced Hg deposition near ASGM likely results from the burning of Hg–gold amalgams,

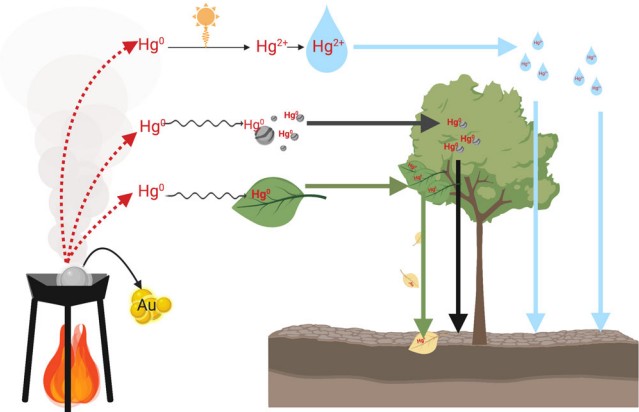

**Fig. 1 Deposition pathways for mercury in the environment in the process of burning a mercury (Hg)–gold (Au) amalgam.** Mercury emitted as gaseous elemental mercury (GEM; Hg$^0$) can undergo three atmospheric pathways to be deposited onto the landscape. First, GEM can be oxidized to ionic Hg (Hg$^{2+}$), which can be entrained in water droplets and deposited as wet or dry deposition to foliar surfaces. Second, GEM can sorb to atmospheric particles (Hg$_p$), which are intercepted by leaves and, along with intercepted ionic Hg, washed onto the landscape via throughfall. Third, GEM can be taken up into leaf tissue, and the Hg deposited onto the landscape as litterfall. Throughfall and litterfall together are considered as an estimate of total Hg deposition. While GEM may also diffuse into and adsorb onto soils and litter directly[77], this is likely not a major pathway for Hg entry into the terrestrial ecosystem.

it is unclear how this Hg is transported across the regional landscape and the relative importance of the different depositional pathways near ASGM.

We expect gaseous elemental Hg concentrations to decline with distance from Hg emission sources. Since two of the three pathways by which Hg is deposited onto the landscape (throughfall and litterfall) are dependent upon Hg interaction with plant surfaces, we would also anticipate the rate of Hg deposition into ecosystems and the risk it poses to animals to be heavily influenced by the structure of vegetation, as suggested by observations in boreal and temperate forests at northern latitudes[23]. However, we also recognize that ASGM activities frequently occur in tropical landscapes, where the canopy structure and relative abundance of exposed leaf area are vastly different. The relative importance of Hg deposition pathways in these ecosystems has yet to be firmly quantified, particularly for forests in proximity to Hg emission sources with an intensity rarely observed in northern forests. In this study, we thus ask: (1) How do gaseous elemental mercury concentrations and depositional pathways vary with proximity to ASGM and leaf area index of regional canopies? (2) Is soil Hg storage related to atmospheric inputs? and (3) Is there evidence that Hg bioaccumulation is elevated in forest-dwelling resident songbirds near ASGM activity? This study is the first to examine Hg depositional inputs near ASGM activity and how canopy cover correlates with these patterns, as well as the first to measure methylmercury (MeHg) concentrations in terrestrial landscapes of the Peruvian Amazon. We measured GEM in the atmosphere, along with total Hg and MeHg in bulk precipitation, throughfall, foliage, litterfall, and soil in both forested and deforested habitats along a 200 km segment of the Madre de Dios River in southeastern Peru. We hypothesized that proximity to ASGM and mining towns where the Hg–gold amalgams are burned would be the most important factors driving atmospheric Hg concentrations (GEM) and wet Hg deposition (bulk precipitation). Because dry Hg deposition (throughfall + litterfall) is related to canopy structure[21,24], we also

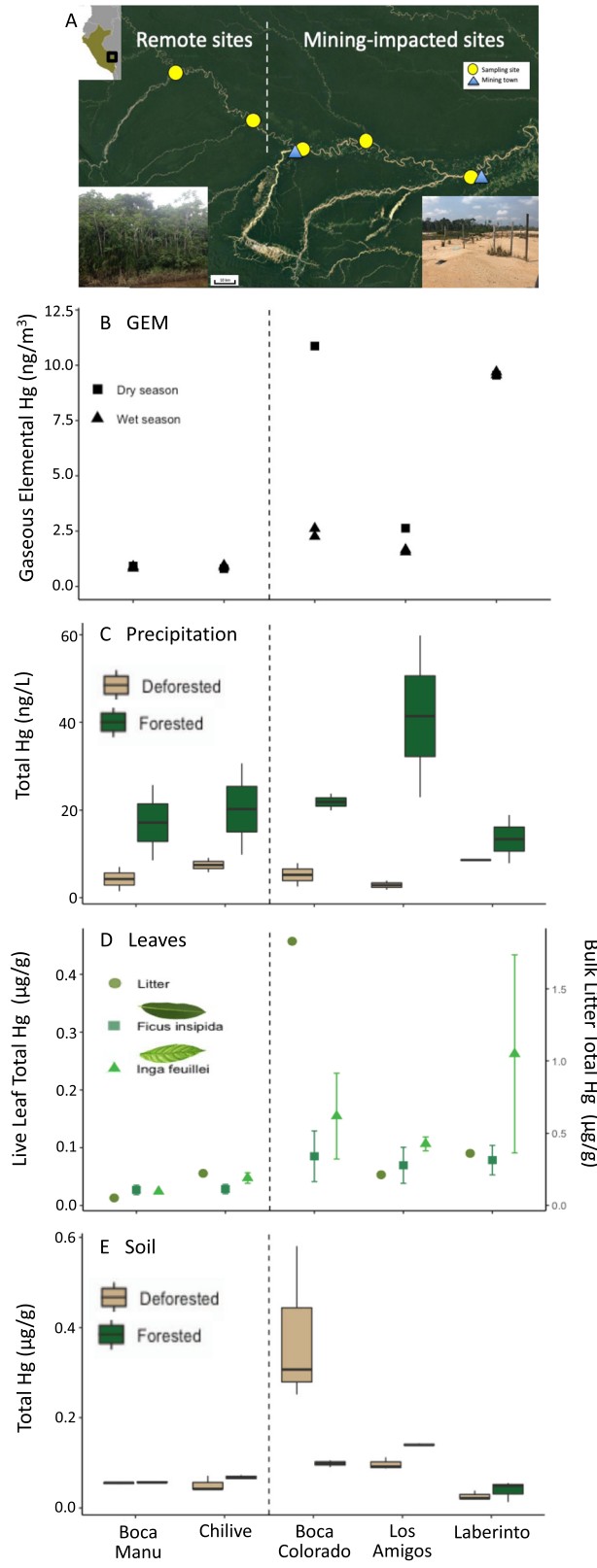

**Fig. 2 Concentrations of mercury in depositing materials and surficial soil in Madre de Dios, Peru. A** Map of the five sampling sites shown as yellow circles. Two sites (Boca Manu, Chilive) are located in areas remote from artisanal gold mining, and three sites (Los Amigos, Boca Colorado, and Laberinto) are located in mining-impacted areas, with mining towns shown as blue triangles. The insets show a typical forested remote site and deforested mining-impacted site. In all figures, the dashed line represents the demarcation between the two remote sites (on the left) and the three mining-impacted sites (on the right). **B** Gaseous elemental mercury (GEM) concentrations at each site for the 2018 dry ($n = 1$ independent sample for each site; square symbol) and wet ($n = 2$ independent samples; square symbol) seasons. **C** Concentration of total mercury in precipitation collected in forested (green boxplots) and deforested (brown boxplots) areas during the 2018 dry season. For all boxplots, the line represents the median value, the box shows Q1 and Q3, and the whiskers denote 1.5 times the interquartile range ($n = 5$ independent samples for each forested site, $n = 4$ independent samples for each deforested site). **D** Concentration of total mercury in leaves collected during the 2018 dry season from the canopy of *Ficus insipida* and *Inga feuillei* (left axis; dark green square and light green triangle symbols, respectively) and as bulk litter on the ground (right axis; olive green circle symbol). Values are shown as mean and standard deviation ($n = 3$ independent samples for live leaves for each site, $n = 1$ independent sample for litter). **E** Concentration of total mercury in surficial soils (top 0–5 cm) collected during the 2018 dry season ($n = 3$ independent samples for each site) in forested (green boxplots) and deforested (brown boxplots) areas. Data for other seasons are shown in Figs. S1 and S2.

## Results and discussion

**Mercury deposition near artisanal gold mining in the Peruvian Amazon.** Our investigation occurred in the southeastern Peruvian Amazon in the department of Madre de Dios, where over 100,000 hectares have been deforested for alluvial ASGM[3] adjacent—and sometimes within—protected land and national reserves. ASGM activity along rivers in this Western Amazonian region has increased dramatically over the past decade[25] and is expected to continue as gold prices remain high and with increased connectivity to urban centers via the Interoceanic Highway[3]. We selected two sites without any mining (Boca Manu and Chilive, approximately 100 and 50 km from ASGM, respectively)—hereafter referred to as "remote sites"—and three sites within the mining zone—hereafter referred to as "mining sites" (Fig. 2A). Two of the mining sites are located in secondary growth forests near the towns of Boca Colorado and Laberinto, and one mining site is located in the intact old-growth forest of the Los Amigos Conservation Concession. Note that the release of Hg vapor from the burning of Hg–gold amalgams regularly occurs within this mining zone, at both the Boca Colorado and Laberinto sites, though exact locations and number of locations are unknown since these activities are generally informal and clandestine; we refer to mining and amalgam burning collectively as "ASGM activity". At each location, we installed deposition samplers both in clearings (deforested areas completely void of woody plants) and beneath the tree canopy (forested areas) in the dry and wet seasons for a total of three seasonal campaigns (each 1–2 months in duration) to collect wet deposition and throughfall, respectively, and deployed passive air samplers in clearings to collect GEM. In the second year, we installed collectors at six additional forested plots at Los Amigos based on the high rates of deposition measured during the first year.

Atmospheric Hg concentrations (GEM) followed our predictions, with high values adjacent to ASGM activity—particularly near the towns where Hg–gold amalgams are burned—and low values in areas distant from active mining (Fig. 2B). At remote

anticipated that forested areas would have higher Hg inputs than neighboring deforested areas, which would be particularly concerning given the high leaf area index and potential for Hg capture in intact Amazonian forests. We further hypothesized that fauna living in forests near mining towns would have higher Hg content than those living in areas far from mining.

locations, GEM concentrations were below the global southern hemisphere average background concentration of approximately 1 ng m$^{-3}$[26]. In contrast, GEM concentrations in all three mining sites were 2−14 times higher than remote sites, with concentrations at the sites near the two mining towns (up to 10.9 ng m$^{-3}$) comparable to, and sometimes exceeding, concentrations from urban and industrial areas of the United States, China, and South Korea[27]. This GEM pattern in Madre de Dios is consistent with Hg–gold amalgam burning as the primary source of elevated atmospheric Hg in this remote Amazonian region.

**Forest canopy as a driver of mercury deposition**. While GEM concentrations in clearings tracked proximity to mining, total Hg concentrations in throughfall were dependent on both proximity to mining and forest canopy structure. This pattern suggests that GEM concentrations alone do not predict where on the landscape elevated Hg will be deposited. We measured the highest concentrations of Hg in throughfall in the intact mature forest within the mining zone (Fig. 2C). The average concentration of total Hg in throughfall at Los Amigos Conservation Concession during the dry season was among the highest reported in the literature (range: 18–61 ng L$^{-1}$), rivaling levels measured in sites contaminated from cinnabar mining and industrial coal combustion in Guizhou, China when considering differences in precipitation volume[28]. These values represent, to our knowledge, the largest measured annual throughfall Hg flux, according to calculations using Hg concentrations and precipitation rates from both the dry and wet seasons (71 µg m$^{-2}$ yr$^{-1}$; Supplementary Table 1). Throughfall total Hg at the other two mining sites was not elevated compared to the remote sites (range: 8−31 ng L$^{-1}$; 22−34 µg m$^{-2}$ yr$^{-1}$). Other than Hg, only aluminum and manganese were elevated in throughfall at the mining sites, which likely is due to land clearing associated with mining; all other measured major and trace elements did not vary between the mining and remote locations (Supplementary Data File 1), a finding consistent with leaf Hg dynamics[29] and ASGM amalgam burning, rather than airborne dust, as the main source of Hg in throughfall.

In addition to serving as sorbents for particulate and gaseous Hg, plant leaves can assimilate GEM directly and incorporate it into tissues[30,31]. Indeed, litterfall was a major source of Hg deposition at sites in close proximity to ASGM activity. Average concentrations of Hg measured in live canopy leaves at all three mining sites (0.080–0.22 µg g$^{-1}$) exceeded published values for temperate, boreal, and alpine forests in North America, Europe, and Asia, as well as other Amazonian forests in South America, located in both remote areas and near point sources[32–34]. Concentrations were comparable to foliar Hg concentrations reported from subtropical mixed forests in China and Atlantic forests in Brazil (Fig. 2D)[32–34]. The highest total Hg concentrations in bulk litter and canopy leaves were measured in the secondary growth forests within the mining zone, following patterns in GEM. However, the estimated litterfall Hg flux was highest in the intact old-growth forest in the mining zone at Los Amigos, presumably due to greater litterfall mass. We estimated the flux of Hg via litterfall at the Los Amigos site to be 66 µg Hg m$^{-2}$ yr$^{-1}$ by taking the measured Hg in litterfall (averaged between the dry and wet seasons) and multiplying by the previously reported litterfall mass for the Peruvian Amazon[35] (Fig. 3A). This input suggests that both proximity to mining and canopy cover are important contributors to Hg loading from ASGM in this region.

Using long term precipitation and litterfall data, we were able to scale our measurements of throughfall and litterfall Hg content derived from three campaigns to perform a preliminary estimate

of total annual atmospheric Hg fluxes (throughfall + litterfall + precipitation) to the Los Amigos Conservation Concession. We found that atmospheric Hg fluxes in forested conservation areas adjacent to ASGM activity are more than 15 times greater than surrounding deforested areas (137 vs. 9 µg Hg m$^{-2}$ yr$^{-1}$; Fig. 3 A, B). This preliminary estimate of Hg loading at Los Amigos exceeds previously reported Hg fluxes in forests of North America and Europe near point sources of Hg (e.g., coal combustion) and is on par with values in industrial China[21,36]. Taken together, approximately 94% of total Hg deposited in conserved forests at Los Amigos occurs via dry deposition (throughfall + litterfall − precipitation Hg), a much higher contribution from dry deposition than most other forested landscapes globally. These results highlight the elevated quantity of Hg from ASGM activity entering forests via dry deposition and the importance of the forest canopy in scavenging ASGM-derived Hg from the atmosphere. We anticipate that observed patterns of highly enriched Hg deposition in forested areas near ASGM activity are not isolated to Peru.

In contrast, deforested areas in the mining zone had lower Hg loading, largely via bulk precipitation with little throughfall and litterfall Hg inputs. Concentrations of total Hg in bulk precipitation within the mining sites were comparable to the values measured at the remote sites (Fig. 2C). Average concentrations of total Hg in dry season bulk precipitation (range: 1.5–9.1 ng L$^{-1}$) were below values previously reported for the Adirondack Mountains of New York[37], and generally below values for remote areas in the Amazon[38]. Thus, in contrast to patterns of GEM, throughfall, and litterfall concentrations at the mining sites, bulk precipitation inputs of Hg are uniformly lower within adjacent deforested areas (8.6–21.5 µg Hg m$^{-2}$ yr$^{-1}$) and do not reflect proximity to mining. Because ASGM requires deforestation[2,3], the cleared areas where mining activity is concentrated receive lower Hg inputs from atmospheric deposition than nearby forested areas, though non-atmospheric direct releases from ASGM such as elemental Hg spillage or tailings can be high[22].

The variation in Hg flux observed in the Peruvian Amazon was driven by large differences within (forested and deforested) and between sites during the dry season (Fig. 2). In contrast, we saw minimal differences within and between sites and low Hg fluxes during the wet season (Supplementary Fig. 1). This seasonal difference (Fig. 2B) likely results from a higher intensity of both mining and dust generation during the dry season. Increased deforestation and the low volume of precipitation during the dry season likely enhances dust generation, thereby increasing the quantity of atmospheric particles which sorb Hg. This production of Hg and dust in the dry season likely lead to patterns in Hg flux within deforested compared to forested areas at the Los Amigos Conservation Concession.

Since Hg inputs from ASGM in the Peruvian Amazon are largely deposited to terrestrial ecosystems via interaction with the forest canopy, we tested whether higher canopy density (i.e., leaf area index) would lead to higher Hg inputs. Within the intact forest at Los Amigos Conservation Concession, we collected throughfall from seven forested plots with different canopy densities. We found that leaf area index is a strong predictor of total Hg inputs via throughfall, with mean total Hg concentrations in throughfall increasing with leaf area index (Fig. 3C). Many other variables also impact Hg inputs via throughfall, including leaf age[34], leaf roughness, stomatal density, wind speed[39], turbulence, temperature, and antecedent dry period.

**Mercury fate in terrestrial ecosystems**. Consistent with the highest rates of Hg deposition, surficial soils (0–5 cm) from the

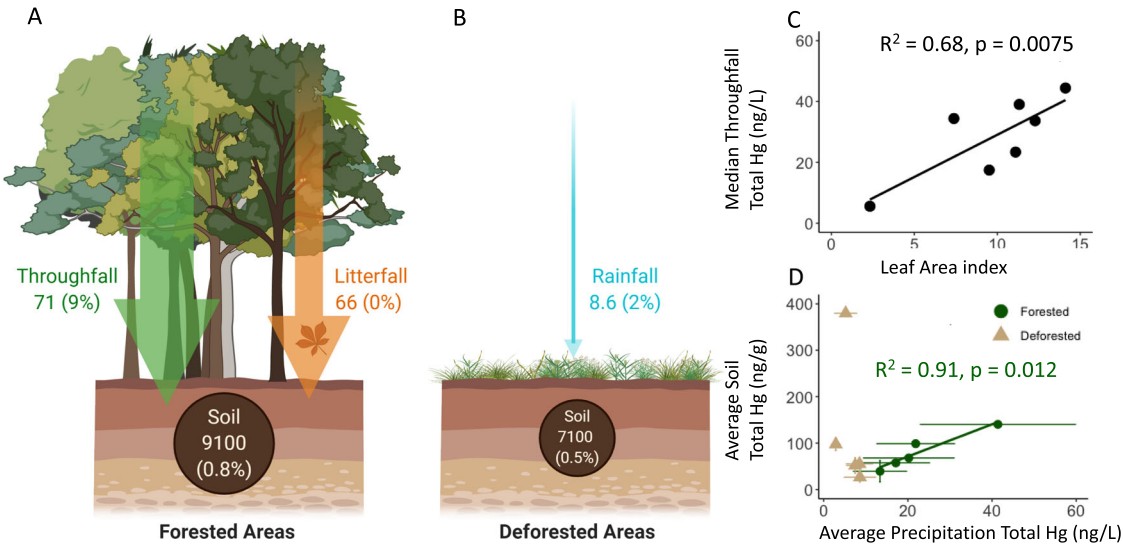

**Fig. 3 Mercury flux and surficial soil pools (0–5 cm) at the Los Amigos Conservation Concession.** Data are shown in **A** forested and **B** deforested areas. The deforested area at Los Amigos represents a clearing for the field station, which makes up a small fraction of total land. Fluxes are shown in arrows and expressed as $\mu g\ m^{-2}\ yr^{-1}$. Pools are shown in circles for the top 0–5 cm of soil and expressed as $\mu g\ m^{-2}$. Percentages represent the percent of the mercury present as methylmercury in the pool or flux. The average concentration between the dry season (2018 and 2019) and wet season (2018) for total mercury in throughfall, bulk precipitation, and litterfall were used for this upscaled estimate of mercury loading. Methylmercury data are based on the 2018 dry season, the only year that it was measured. For information on pool and flux calculations, see the "Methods". **C** Relationship between total mercury concentration in throughfall and leaf area index at the eight plots at Los Amigos Conservation Concession according to an ordinary least square regression. **D** Relationship between precipitation total mercury concentration and surficial soil mercury total concentration at all five sites in the forested (green circle) and deforested (brown triangle) areas according to an ordinary least square regression (error bars show standard deviation).

Los Amigos forested site had the highest total Hg concentrations ($140\ ng\ g^{-1}$ in the 2018 dry season; Fig. 2E) of our study sites. Moreover, Hg concentrations were enriched throughout the measured vertical soil profile (range of $138–155\ ng\ g^{-1}$ up to 45 cm in depth; Supplementary Fig. 3). The only site to exhibit higher surficial soil Hg concentrations within the 2018 dry season was a deforested site near a mining town (Boca Colorado). At this site, we hypothesize that the extremely high concentrations may have been due to local contamination of elemental Hg during the amalgamation process, as concentrations were not elevated at depth (>5 cm). It is also likely that the fraction of atmospheric Hg deposition lost by evasion (i.e., release of Hg to the atmosphere) from soil is considerably lower in forested areas due to canopy cover compared to deforested areas[40], suggesting that a sizable fraction of Hg deposited to conservation areas is retained within the soil. The soil total Hg pool in the intact forests at Los Amigos Conservation Concession was $9100\ \mu g\ Hg\ m^{-2}$ within the top five cm and over $80,000\ \mu g\ Hg\ m^{-2}$ within the top 45 cm.

Since foliage incorporates Hg predominantly from the atmosphere and not from the soil[30,31] and then delivers this Hg to the soil via throughfall, the high rates of Hg in deposition are likely driving the observed patterns in soils. We found a strong correlation between average total Hg concentrations in surficial soil and total Hg concentrations in throughfall across all forested sites, and no relationship between Hg in surficial soil and total Hg concentrations in bulk precipitation in deforested areas (Fig. 3D). Similar patterns are also evident for the relationship between surficial soil Hg pool and throughfall total Hg flux in forested, but not deforested areas (surficial soil Hg pool vs. bulk precipitation total Hg flux).

Nearly all research on terrestrial Hg pollution associated with ASGM has been limited to measurements of total Hg, yet it is MeHg concentrations that determine Hg bioavailability and subsequent trophic accumulation and exposure. In terrestrial ecosystems, Hg is methylated by microorganisms under anoxic conditions[41,42], and thus it is often assumed that MeHg

concentrations are low in upland soils. However, we document for the first time that there are measurable MeHg concentrations within Amazonian soils near ASGM, suggesting that elevated MeHg concentrations extend beyond aquatic ecosystems and into terrestrial environments within these ASGM-impacted areas, including soils that are inundated during the wet season as well as those that remain dry throughout the year. The highest surficial soil concentrations of MeHg during the 2018 dry season occurred at two of the forested sites in the mining zone (Boca Colorado and Los Amigos Conservation Concession; $1.4\ ng\ MeHg\ g^{-1}$, 1.4% Hg as MeHg and $1.1\ ng\ MeHg\ g^{-1}$, 0.79% Hg as MeHg, respectively). As these percentages of Hg present as MeHg are comparable to other terrestrial sites around the globe (Supplementary Figure 4), it appears that the high concentrations of MeHg are due to high total Hg inputs and high storage of total Hg in soil, rather than efficient net conversion of inorganic Hg to MeHg (Supplementary Fig. 5). Our results represent the first measurements of MeHg in soils near ASGM in the Peruvian Amazon. Based on other studies that report higher MeHg production in flooded versus dry landscapes[43,44], we expect that MeHg concentrations will be even higher in nearby forested seasonal and permanent wetlands experiencing similar Hg loadings. Though it remains to be determined if MeHg poses a toxicity risk for terrestrial wildlife near gold mining activity, these forests near ASGM activity could be hotspots for Hg bioaccumulation into terrestrial food webs.

**Implications for tropical forests and biodiversity**. The most important and novel implication of our work is the documentation of elevated quantities of Hg being delivered to forests near ASGM activity. Our data show that this Hg is available to, and moving through, terrestrial food webs. Moreover, very large quantities of Hg are stored in biomass and soils with the potential for release with land use change[4] and forest fires[45,46]. The southeastern Peruvian Amazon is one of the most biodiverse

ecosystems on the planet for vertebrate and insect taxa[47]. The high structural complexity within intact old-growth tropical forests promotes bird biodiversity[48] and provides niches for a wide range of forest-dwelling species[49]. For this reason, over 50% of the Madre de Dios region is designated as either protected land or national reserve[50]. International pressure to control illegal ASGM activity within the conservation buffer zone of Tambopata National Reserve has grown significantly over the last ten years, resulting in a major enforcement action by the Peruvian government in 2019 (Operación Mercurio). Yet, our results suggest that the very forest complexity that is the basis for Amazonian biodiversity makes this region highly vulnerable to enhanced Hg loading and storage on the landscape from ASGM-related Hg emissions, leading to the highest ever reported measurements of throughfall Hg flux globally and elevated litterfall Hg flux in intact forests near ASGM, according to our preliminary estimates. While our investigation occurred in a protected forest, the pattern of elevated Hg inputs and retention would apply to any old growth primary forest near ASGM activity including buffer zones, making these results relevant to protected and non-protected forests alike. The risk to the landscape of Hg from ASGM is, therefore, a function not only of direct Hg inputs via atmospheric emissions, spills, and tailings, but also of the landscape's potential to capture, store, and transform Hg into the more bioavailable form of MeHg, suggesting differential impacts for the global Hg pool and terrestrial wildlife depending upon forest cover near mining.

By sequestering atmospheric Hg, intact forests near ASGM may reduce the risk of Hg to nearby aquatic ecosystems and to the global atmospheric Hg pool. If these forests are cleared for the expansion of mining or agricultural activities, legacy Hg could be mobilized from the terrestrial to aquatic ecosystem via forest fires, evasion, and/or runoff[45,46,51–53]. In the Peruvian Amazon, where ~180 tons of Hg are used annually in ASGM[54] and approximately a quarter of this Hg is emitted into the atmosphere[55], 30 million hectares of intact forested land would be required to capture all this Hg given the rates observed at the Los Amigos Conservation Concession. This is an area about 7.5 times that of the total extent of protected lands and natural reserves in the Madre de Dios region (~4 million ha), a department that has the largest fraction of land in protected status of any other Peruvian department, and much of this intact forested land is not within the depositional radius of Hg from ASGM. Forest sequestration of Hg in intact forests is therefore not sufficient to prevent ASGM-derived Hg from entering the regional and global atmospheric Hg pool, suggesting the importance of reducing releases of Hg from ASGM. The fate of the large amount of Hg that is stored within terrestrial systems is greatly influenced by conservation policies. Future decisions regarding how intact forests are managed, particularly in areas adjacent to ASGM activity, thus have implications for Hg mobilization and bioavailability now and for decades to come.

Even if forests could sequester all the Hg released in tropical forests, this is not a panacea for Hg pollution because terrestrial food webs may also be vulnerable to Hg exposure. We know little about the concentration of Hg in biota within these intact forests, but these first measurements of terrestrial Hg deposition and soil MeHg suggest that high Hg loading and elevated MeHg within soils could increase the risk of exposure to high trophic level consumers inhabiting these forests. Data from previous studies on terrestrial Hg bioaccumulation in temperate forests found that bird blood Hg concentrations were correlated with Hg concentrations in deposition and that songbirds consuming entirely terrestrially-derived food can exhibit elevated Hg concentrations[56,57]. Elevated Hg exposure in songbirds leads to reduced reproductive performance and success, decreased

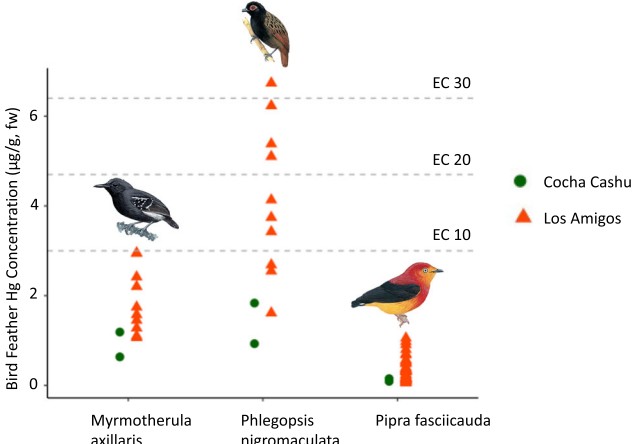

**Fig. 4 Total mercury concentrations in tail feathers of bird species in the Peruvian Amazon.** Data were collected at the Los Amigos Conservation Concession (n = 10 for *Myrmotherula axillaris* [understory invertivore] and *Phlegopsi nigromaculata* [ant-following invertivore], n = 46 for *Pipra fasciicauda* [frugivore]; red triangle symbol) and the remote site of Cocha Cashu Biological Station (n = 2 for each species; green circle symbol). The effective concentrations (EC) at which reproductive success is reduced by 10, 20, and 30% (see Evers[58]) are shown. Bird photos are modified from Schulenberg[65].

offspring survival, impaired development, altered behavior, physiological stress, and mortality[58,59]. If this pattern holds for the Peruvian Amazon, the high Hg flux occurring in intact forests could lead to high Hg concentrations in birds and other biota and potentially to adverse effects. This is of particular concern since this region is a global biodiversity hotspot[60]. These results highlight the importance of preventing ASGM activity from occurring within national reserves and the buffer zones that surround them. Formalizing ASGM activity[15,16] could be a mechanism for ensuring protected lands are not mined.

To assess whether the Hg deposited into these forested areas is entering terrestrial food webs, we measured total Hg concentrations in the tail feathers of several species of resident songbirds from the Los Amigos Conservation Concession (mining-impacted) and Cocha Cashu Biological Station (unimpacted old growth forest), a remote site 140 km beyond our most upstream sampling site at Boca Manu. For all three species for which multiple individuals were sampled at each site, Hg was elevated in the birds from Los Amigos compared with Cocha Cashu (Fig. 4). This pattern was present regardless of feeding habits, as our samples included the understory invertivores *Myrmotherula axillaris*, ant-following invertivores *Phlegopsis nigromaculata*, and frugivores *Pipra fasciicauda* (1.8 [n = 10] vs. 0.9 µg g$^{-1}$ [n = 2], 4.1 [n = 10] vs. 1.4 µg g$^{-1}$ [n = 2], and 0.3 [n = 46] vs. 0.1 µg g$^{-1}$ [n = 2], respectively). Of the ten *Phlegopsis nigromaculata* individuals sampled at Los Amigos, three exceeded the EC10 (effective concentration at which reproductive success is decreased by 10%), three exceeded the EC20, and one exceeded the EC30 (see EC standards in Evers[58]), while no individuals of any species at Cocha Cashu exceeded the EC10. These initial findings of average Hg concentrations 2−3 times higher and individual Hg concentrations up to twelve times higher in songbirds from a protected forest adjacent to ASGM activity raise considerable concern about the extent to which Hg pollution from ASGM may be entering terrestrial food webs. These results highlight the importance of preventing ASGM activity from occurring within national parks and the buffer zones that surround them.

The extent of ASGM in the Peruvian Amazon has increased over 40% in protected areas since 2012 and even more in unprotected areas[2,25]. The continued use of Hg in ASGM could have devasting impacts on wildlife inhabiting these forests. Even if miners eliminated the use of Hg immediately, this contaminant has a legacy in soils that can extend for centuries, with the potential for elevated losses associated with deforestation and forest fires[61,62]. Mercury contamination from ASGM thus could have long lasting impacts on biota of intact forests near ASGM, with both the current risks and the potential for future contamination through Hg liberation and remobilization maximized in old growth forests with the highest conservation value. Our finding that terrestrial biota may be at considerable risk from ASGM-derived Hg pollution should provide further incentive for on-going efforts to reduce the release of Hg from ASGM. Those efforts include a variety of approaches that range from the relatively simple Hg-capture system of retorts to the more challenging economic and social investments that would formalize this activity and reduce the financial incentives of conducting ASGM illegally.

## Methods

**Sample collection.** We established five sites along a 200 km reach of the Madre de Dios River. We chose sampling locations based on their proximity to intense ASGM activities, with approximately 50 km between each sampling site and accessible by the Madre de Dios River (Fig. 2A). We selected two sites without any mining (Boca Manu and Chilive, approximately 100 and 50 km from ASGM, respectively), hereafter referred to as "remote sites". We selected three sites within the mining zone, hereafter referred to as "mining sites", with two of the mining sites located in secondary growth forests near the towns of Boca Colorado and Laberinto and one mining site located in the intact old-growth forests of the Los Amigos Conservation Concession. Note that the release of Hg vapor from the burning of Hg–gold amalgams regularly occurs within this mining zone, at both the Boca Colorado and Laberinto sites, though exact locations and number of locations are unknown since these activities are generally illegal and clandestine; we refer to mining and amalgam burning collectively as "ASGM activity". At each of the five sites, we installed deposition samplers both in clearings (deforested areas completely void of woody plants) and beneath the tree canopy (forested areas) in the 2018 dry season (July and August 2018) and 2018 wet season (December 2018 and January 2019) to collect wet deposition ($n = 3$) and throughfall ($n = 4$), respectively. Precipitation samples were collected over the course of four weeks in the dry season and two to three weeks in the wet season. In the second year of sampling during the dry season (July and August 2019), we installed collectors ($n = 4$) at six additional forested plots at Los Amigos for five weeks based on the high rates of deposition measured during the first year, for a total of seven forested plots and one deforested plot at Los Amigos. The distance between plots ranged from 0.1 to 2.5 km. We collected a GPS waypoint at each plot using a handheld Garmin GPS.

We deployed passive air samplers (PAS) for Hg at each of the five sites during the 2018 dry season for a two-month period (July–August 2018) and the 2018 wet season for one month (December 2018−January 2019). One PAS sampler was deployed per site during the dry season, and PAS samplers were deployed in duplicate during the wet season. The PAS (developed by McLagan et al.[63]) collects gaseous elemental mercury (GEM) by passive diffusion through a Radiello© diffusive barrier and sorption onto a sulfur-impregnated carbon sorbent (HGR-AC). The diffusion barrier of the PAS acts as a barrier to prevent the passage of gaseous organic Hg species; thus, only GEM is sorbed onto the carbon[64]. We attached PAS to posts approximately 1 m above the ground using plastic cable ties. All samplers were sealed with parafilm or stored in double resealable plastic bags prior to and post deployment. We collected field blanks and trip blank PASs to assess contamination introduced during sampling, storage in the field, storage in the laboratory, and during sample transport.

During the deployment periods at all five sampling sites, we placed three precipitation collectors for Hg analysis and two collectors for other chemical analyses in the deforested sites and four throughfall collectors for Hg analysis, and two collectors for other chemical analyses in the forested sites. Collectors were placed within one meter of each other. Note that while we installed a consistent number of collectors at each site, during some collection periods we had a smaller sample size due to flooding of sites, human interference with collectors, and connection malfunction between the tubing and collector bottle. At each forested and each deforested site, one of the collectors for Hg analysis contained a 500 mL bottle, while the others contained a 250 mL bottle; all collectors for other chemical analyses contained a 250 mL bottle. These samples were stored cold until access to a freezer allowed them to be frozen, transported to the United States on ice, and stored frozen until analyzed. Collectors for Hg analysis consisted of a glass funnel connected to a new polyethylene terephthalate copolyester glycol (PETG) bottle via

new styrene-ethylene-butadiene-styrene block polymer (C-Flex) tubing with a loop as a vapor lock. At the time of deployment, all 250 mL PETG bottles were acidified with 1 mL of trace metal grade hydrochloric acid (HCl), and all 500 mL PETG bottles were acidified with 2 mL of trace metal grade HCl. Collectors for other chemical analyses consisted of a plastic funnel connected to a polyethylene bottle via new C-Flex tubing with a loop as a vapor lock. Prior to deployment, all glass funnels, plastic funnels, and polyethylene bottles were acid washed. We collected samples using the clean hands-dirty hands protocol (EPA Method 1669), kept the samples as cold as possible until return to the United States, and then stored samples at 4 °C until analysis. A previous study using this methodology has shown laboratory blanks below the detection limit and standard spikes to have recoveries of 90−110%[37].

At each of the five sites, we collected foliage as canopy leaves, grab leaf samples, fresh litterfall, and bulk litter using clean hands-dirty hands protocol (EPA Method 1669). All samples were collected under a collection permit from SERFOR in Peru and imported to the United States under a USDA import permit. We collected canopy leaves from two tree species found at all sites: an emergent tree species (*Ficus insipida*) and a medium-sized tree (*Inga feuilleei*). We collected leaves from the canopy of the trees ($n = 3$ for each species) using a Notch Big Shot slingshot in the 2018 dry season, 2018 wet season, and 2019 dry season. We collected grab leaf samples ($n = 1$) by sampling leaves in each plot from tree branches less than two meters from the ground in the 2018 dry season, 2018 wet season, and 2018 dry season. In 2019, we also collected grab leaf samples ($n = 1$) from the additional six forested plots at Los Amigos. We collected fresh litterfall ("bulk litter") in plastic mesh-lined baskets ($n = 5$) in the 2018 wet season at all five forested sites and the 2019 dry season at the Los Amigos plots ($n = 5$). Note that while we installed a consistent number of baskets at each site, during collection periods we had a smaller sample size due to flooding of sites and human interference with collectors. All litterfall baskets were placed within one meter of the precipitation collectors. We collected bulk litter as grab samples of litterfall on the ground in the 2018 dry season, 2018 wet season, and 2019 dry season. In the 2019 dry season, we also collected bulk litter in all Los Amigos plots. We cold-stored all leaf samples until access to a freezer allowed them to be frozen, transported to the United States on ice, then stored frozen until processed.

We collected soil samples in triplicate ($n = 3$) from all five sites (open and canopy) during all three seasonal campaigns and from the Los Amigos plots in the 2019 dry season. All soil samples were collected within one meter of the precipitation collectors. We collected soil samples as surficial soil under the litter layer (0–5 cm) using a soil corer. Additionally, in the 2018 dry season, we collected soil cores up to 45 cm in depth and divided them into five depth segments. At Laberinto, we were only able to collect one soil profile because the water table was close to the soil surface. We collected all samples using the clean hands-dirty hands protocol (EPA Method 1669). We cold-stored all soil samples until access to a freezer allowed them to be frozen, transported to the United States on ice, then stored frozen until processed.

Birds were captured using mist nests set up at both dawn and dusk, during the coolest parts of the day. At the Los Amigos Conservation Concession, we placed five mist nests ($1.8 \times 2.4$) at nine locations. At Cocha Cashu Biological Station, we placed eight to ten mist nests ($12 \times 3.2$ m) at nineteen locations. At both locations, we collected the first central tail feather from each bird, or if not available, the next oldest feather. We stored the feathers in clean Ziploc bags or manilla envelopes with silica gel. We assembled a photographic record and morphological measurements to identify the species based on Schulenberg[65]. Both studies were supported by permits from SERFOR as well as the Animal Studies Committees (IACUC). When comparing bird feather Hg concentrations, we examined those species for which feathers had been collected at both Los Amigos Conservation Concession and Cocha Cashu Biological Station (*Myrmotherula axillaris, Phlegopsis nigromaculata, Pipra fasciicauda*).

To determine leaf area index (LAI), Lidar data were collected using the GatorEye Uninhabited Flying Laboratory, which is a sensor fusion drone system (details available at www.gatoreye.org, with plot scale data download also available using the "2019 Peru Los Amigos June" link)[66]. Lidar was collected at the Los Amigos Conservation Concession in June 2019 at 80 m aboveground level, 12 m/s flight speed, and with adjacent flightlines 100 m apart, resulting in a 75% sidelap coverage percentage. Point density exceeded 200 points per m$^2$ distributed across the vertical forest profile. The flight area overlapped all sampling plots at Los Amigos for the 2019 dry season.

**Laboratory analyses.** We quantified total Hg concentrations of GEM collected by PAS by thermal desorption, amalgamation, and atomic absorption spectroscopy (USEPA Method 7473) using a Hydra C instrument (Teledyne, CV-AAS). We performed calibration of the CV-AAS using National Institute of Standards and Technology (NIST) standard reference material 3133 (Hg standard solution, 10.004 mg g$^{-1}$), with a detection limit of 0.5 ng Hg. We performed continuous calibration verification (CCV) using NIST SRM 3133 and quality control standard (QCS) using NIST 1632e (bituminous coal, 135.1 mg g$^{-1}$). We divided each sample into separate boats, placed it between two thin layers of sodium carbonate ($Na_2CO_3$) powder, and covered it with a thin layer of aluminum hydroxide ($Al(OH)_3$) powder[67]. We measured the entire HGR-AC contents from each sample to remove any inhomogeneity in the distribution of Hg within the HGR-AC

sorbent. Therefore, we calculated the Hg concentration for each sample based on the sum of total Hg measured for each boat and the entire HGR-AC sorbent contents in the PAS. Given that only one PAS sample was collected in the 2018 dry season from each site for concentration measurements, method quality control and assurance were carried out by bracketing samples with monitoring procedural blanks, internal standards, and matrix-matched standards. In the 2018 wet season, we measured PAS samples in duplicate. Values were deemed acceptable when the relative percent difference (RPD) measured for both CCV and matrix-matched standards were within 5% of the accepted values, and all procedural blanks were below detection limit (BDL). We blank-corrected measured total Hg in PAS using concentrations determined from field and trip blanks ($0.81 \pm 0.18$ ng g$^{-1}$, $n = 5$). We calculated GEM concentrations using the total mass of blank-corrected sorbed Hg divided by the deployment time and sampling rate (volume of air stripped of gaseous Hg per unit of time; 0.135 m$^3$ day$^{-1}$)[63,68] adjusted for temperature and wind using average temperature and wind measurements for the Madre de Dios region as obtained from World Weather Online[68]. The reported standard error of measured GEM concentrations is based on the error of external standards ran before and after the samples.

We analyzed water samples for total Hg via oxidation with bromine chloride for a minimum of 24 h, followed by stannous chloride reduction and analysis with purge and trap, cold vapor atomic fluorescence spectroscopy (CVAFS), and gas chromatographic (GC) separation (EPA Method 1631, revision E) on a Tekran 2600 Automated Total Mercury Analyzer. We performed CCV for the 2018 dry season samples using Ultra Scientific certified aqueous Hg standard (10 µg L$^{-1}$) and initial calibration verification (ICV) using NIST certified reference material 1641D (mercury in water, 1.557 mg kg$^{-1}$), with a detection limit of 0.02 ng L$^{-1}$. For the 2018 wet season and 2019 dry season samples, we performed calibration and CCV using Brooks Rand Instruments Total Mercury Standard (1.0 ng L$^{-1}$), and ICV using SPEX Centriprep Inductively Coupled Plasma Mass Spectrometry (ICP-MS) Multi-Element in Solution Standard 2 A, with a detection limit of 0.5 ng L$^{-1}$. All standards had recoveries within 15% of the accepted values. The field blank, digestion blanks, and analysis blanks were BDL.

We lyophilized soil and leaf samples for five days. We homogenized the samples and then analyzed them for total Hg on a Milestone Direct Mercury Analyzer (DMA-80) via thermal decomposition, catalytic reduction, amalgamation, desorption, and atomic absorption spectroscopy (EPA Method 7473). For the 2018 dry season samples, we performed calibration of the DMA-80 using NIST 1633c (coal fly ash, 1005 ng g$^{-1}$) and Canadian National Research Council certified reference material MESS-3 (marine sediment, 91 ng g$^{-1}$). We performed CCV and MS using NIST 1633c and QCS using MESS-3, with a detection limit of 0.2 ng Hg. For the 2018 wet season and 2019 dry season samples, we performed calibration of the DMA-80 using Brooks Rand Instruments Total Mercury Standard (1.0 ng L$^{-1}$). We performed CCV and MS using NIST standard reference material 2709a (San Joaquin Soil, 1100 ng g$^{-1}$) and QCS using DORM-4 (fish protein, 410 ng g$^{-1}$), with a detection limit of 0.5 ng Hg. For all seasons, we analyzed all samples in duplicate and accepted values when the RPD between the two samples was within 10%. All standards and matrix spikes had average recoveries within 10% of the accepted values, and all blanks were BDL. All reported concentrations are for dry mass.

We analyzed water samples from all three seasonal campaigns, leaf samples from the 2018 dry season, and soil samples from all three seasonal campaigns for MeHg. We extracted water samples with trace grade sulfuric acid for a minimum of 24 h[69], digested leaves with 2% potassium hydroxide in methanol at 55 °C for a minimum of 48 h[70], and digested soils via microwave digestion with trace metal grade HNO$_3$ acid[71,72]. We analyzed the 2018 dry season samples via aqueous ethylation with sodium tetraethylborate, purge and trap, and CVAFS on a Tekran 2500 spectrometer (EPA Method 1630). We performed calibration and CCV using Frontier Geosciences certified laboratory MeHg standards and QCS for sediment using ERM CC580, with a method detection limit of 0.2 ng L$^{-1}$. We analyzed the 2019 dry season samples by aqueous ethylation with sodium tetraethylborate, purge and trap, CVAFS, GC, and ICP-MS on an Agilent 770 (EPA Method 1630)[73]. We performed calibration and CCV using Brooks Rand Instruments Methylmercury Standard (1 ng L$^{-1}$), with a method detection limit of 1 pg. For all seasons, all standards had recoveries within 15% of the accepted values, and all blanks were BDL.

We analyzed bird feathers for total Hg on a Milestone Direct Mercury Analyzer (DMA-80) via thermal decomposition, catalytic reduction, amalgamation, desorption, and atomic absorption spectroscopy (EPA Method 7473) at the Biodiversity Research Institute Toxicology Lab (Portland, ME, USA), with a method detection limit of 0.001 µg g$^{-1}$. We performed calibration of the DMA-80 with DOLT-5 (dogfish liver, 0.44 µg g$^{-1}$), CE-464 (5.24 µg g$^{-1}$), and NIST 2710a (Montana soil, 9.888 µg g$^{-1}$). We performed CCV and QCS using DOLT-5 and CE-464. All standards had average recoveries within 5% of the accepted values, and all blanks were BDL. All duplicates were within 15% RPD. All reported concentrations for feather total Hg are for fresh weight (fw).

We filtered water samples for other chemical analyses with a 0.45 µm membrane filter. We analyzed water samples for anions (chloride, nitrate, sulfate) and cations (calcium, magnesium, potassium, sodium) via ion chromatography (EPA Method 4110B) [USEPA, 2017a] with a Dionex ICS 2000 ion chromatograph. All standards had recoveries within 10% of the accepted values, and all blanks were BDL. We analyzed water samples for trace elements via inductively coupled plasma mass spectrometry with a Thermofisher X-Series II. Instrument calibration

standards were prepared via serial dilution of certified water standard NIST 1643f. All blanks were BDL.

**Data analyses and statistical analyses.** All fluxes and pools reported in the text and figures use average concentration values for the dry and wet seasons. For estimates of pools and fluxes using minimum and maximum measured concentrations during the dry and wet seasons (averaged together for the two seasons for an annual flux), see Supplementary Table 1. We calculated the forest Hg flux at Los Amigos Conservation Concession as the sum of Hg inputs via throughfall and litterfall. We calculated the deforested Hg flux from bulk precipitation Hg deposition. We calculated the average cumulative annual rainfall over the past decade (2009–2018) as approximately 2500 mm yr$^{-1}$ using daily rainfall measurements from Los Amigos (collected as part of EBLA, available from ACCA upon request). Note that in the calendar year of 2018, annual rainfall was close to this average (2468 mm), while the wettest months (January, February, and December) accounted for approximately half of the annual precipitation (1288 mm of the 2468 mm total). We therefore use the average of wet and dry season concentrations for all flux and pool calculations. This also allows us to not only account for differences in precipitation quantity between the wet and dry seasons, but also for differences in the extent of ASGM activity between these two seasons. Since reported literature values of annual Hg fluxes in tropical forests vary between scaling up Hg concentrations from both the dry and wet seasons or from only the dry season, when comparing our calculated flux to literature values, we directly compare our calculated Hg flux when the other study collected samples in both the dry and wet seasons and re-estimate our flux using only dry season Hg concentrations when the other study collected samples only in the dry season (e.g.,[74]).

To determine annual total Hg loading in throughfall, bulk precipitation, and litterfall at Los Amigos, we used the average total Hg concentration between the dry season (average for 2018 and 2019 at all Los Amigos sites) and wet season (average for 2018). For total Hg loading at other sites, the average concentration between the 2018 dry season and 2018 wet season was used. For MeHg loading, we used data from the 2018 dry season, the only year that MeHg was measured. To estimate litterfall Hg flux, we used literature estimates of 417 g m$^{-2}$ yr$^{-1}$ litterfall rate in the Peruvian Amazon and concentrations of Hg collected from leaves in the litterbaskets[35]. For soil Hg pools in the upper 5 cm of the soil, we used measured soil total Hg (2018 and 2019 dry season, 2018 wet season) and MeHg concentrations from the 2018 dry season with a bulk density estimate of 1.25 g cm$^{-3}$ from the Brazilian Amazon[75]. We were only able to make these budget calculations at our primary study site of Los Amigos where datasets for long-term rainfall are available and the intact forest structure allows for the use of previously collected litterfall estimates.

We processed Lidar flightlines using the GatorEye multiscale postprocessing workflow, which automatically calculates cleaned merged point clouds and raster products, including a digital elevation model (DEM) at $0.5 \times 0.5$ m resolution. We used the DEM and the cleaned Lidar point cloud (WGS-84, UTM 19S Meters) as inputs into the GatorEye Leaf Area Density (G-LAD) workflow, which calculates calibrated leaf area estimates (m$^2$) per voxel (m$^3$) from the ground through the top of the canopy at a resolution of $1 \times 1 \times 1$ m, as well as the derived LAI (the sum of LAD within each $1 \times 1$ m vertical column). The LAI value for each plot GPS point was then extracted.

We performed all statistical analyses using R version 3.6.1 statistical software[76], and we made all visualizations using ggplot2. We performed statistical tests using an alpha of 0.05. Relationships between two quantitative variables were evaluated using ordinary least square regression. We performed comparisons between sites using the non-parametric Kruskal test followed by the pairwise Wilcox test.

## Data availability

All data included in this manuscript are available in Supplementary Information and in the associated data paper[78]. Precipitation data are available from Conservación Amazónica (ACCA) upon request.

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

## Acknowledgements

We thank Ramiro Cordova Salas, Francisco Phuno Soncco, Bryan Huamantupa Rivera, Cecilio Huamantupa, Arianna Basto, Kelsey Lansdale, Melissa Marchese, Arabella Chen, Christian Lara, Annie Lee, Fernanda Machicao, Tatiana Manidis, Laura Naslund, Centro de Innovación Científica Amazónica (CINCIA), Conservación Amazónico (ACCA), Los Amigos Conservation Concession, and especially our Peruvian technicians for field assistance; Brooke Hassett, Kim Hutchison, Gary Dwyer, Nelson Rivera, Faye Koenigs-mark, Natalia Neal-Walthall, Austin Wadle, Rachel Coyte, Rand Alotaibi, and Arianna Agostini for laboratory assistance; Jennifer Swenson for remote sensing advice; and Marx Gomez-Liendo for translating the manuscript into Spanish. Funding was provided to JRG by Duke Global Health Institute Dissertation Fieldwork Grant, Duke Global Health Institute Doctoral Scholar Program, Duke University Bass Connections, Duke University Center for Latin American and Caribbean Studies Tinker Research Travel Grant Award, Duke University Center for International and Global Studies Research and Training Grant, Duke University Dissertation Research International Travel Award, Geological Society of America Grants in Aid of Research, Lewis and Clark Fund for Exploration and Field Research, and National Science Foundation Graduate Research Fellowship. Funding was provided to ESB by Josiah Charles Trent Memorial Foundation Endowment Fund Grant. This manuscript is also available in Spanish and can be found in the Supplementary Material available online.

## Author contributions

J.R.G. contributed to conception, study design, data collection, data analysis, and manuscript preparation. N.S. contributed to conception, data collection, and data analysis. E.A.U. contributed to data analysis. E.B., G.E., G.I., K.L., M.J.M., A.M., C.M., R.P.P., V.S., and M.W. contributed to data collection and data analysis. E.S.B. contributed to conception and study design. A.A.Z., B.B., C.T.D., D.C.E., L.E.F., H.H., W.P., M.S., and C.V. provided analytical lab and/or field site logistic support. All authors provided manuscript feedback.

## Competing interests

The authors declare no competing interests.
