## [Peer Review File · Nature Communications]

REVIEWER COMMENTS

Reviewer #1 (Remarks to the Author):

This article addresses an issue of major concern within the field on environmental mercury studies. Unfortunately it seems that ASGM is increasing and unlikely to diminish in the near future. The fact that according to the most recent emission inventories it has overtaken power production from fossil fuels, as the authors point out, has effectively shifted the major emission source of atmospheric mercury from the Global North to the Global South, which the authors also point out, but may want to emphasize a little more in the Abstract. This shift also marks a move from the major mercury emission sources being located at mid-latitudes to more tropical regions, with consequences for Hg dispersion and deposition patterns.

This manuscript documents detailed measurements of mercury in a tropical ecosystem close to ASGM activities, it is a significant contribution to the field of environmental mercury cycling, and addresses a pressing problem in the field.

The article is well written and clearly presented, and I have only a few pedantic and stylistic points to make, I thoroughly recommend publication with very minor revision.

In the paragraph beginning on line 78, regarding the burning off of Hg from amalgam. This is often done without a retort to recover Hg, however efforts from the UNEP global mercury partnership, UNIDO and a number of NGOs are making efforts to encourage miners to improve practices, and I think it would be good to mention these efforts somehow.

Line 87. I don't know if it's the journal's policy, but 'see summary table in 16' is horrible. Is it not possible to write 'see summary table in Gersoin et al., (16)'? This occurs in a few other places in the manuscript.

Line 108. It may not be clear to readers why throughfall, which occurs when precipitation effectively washes previously deposited Hg from the canopy and deposits it to the forest floor is considered to be dry deposition, it may be an idea to cite the Wright paper here (ref 28) and add a short comment summarising the description in the reference "Mercury in throughfall includes wet-deposited Hg above the canopy and a portion of dry-deposited Hg washed off from the canopy. Thus, litterfall Hg likely represents the low end of Hg dry deposition when Hg emission from the underlying soil is small, although it can be higher than the actual dry deposition above the canopy due to the interception of emitted Hg by forest leaves if soil Hg emissions are high and the ambient Hg concentrations above the forests are low. The difference between throughfall and open-area wet deposition, with the former collected inside and the latter outside a forest, should also represent a portion of the dry deposition. Thus, on annual basis, dry deposition can be approximated as the sum of litterfall and throughfall Hg minus wet-deposited Hg." This is mentioned later, line 188, but I think it would be useful to clarify the concept with the reader at the start.

Line 116. Not everybody knows what placer deposits are, a short description would avoid the reader skipping to wikipedia whilst reading (the same probably applies to neotropical, but it's in the abstract so perhaps there's a word limit).

Line 122. Are the 'remote' sites upwind of where ASGM activity is occurring, is that why they can be considered remote? 50km downwind would seem relatively close to me.

A large part of this first paragraph of the Results and Discussion is repeated pretty much word for word in the Methods (Sample Collection) section. Probably one could make reference to the other.

Line 151. The concentration of total Hg in throughfall will be in part determined by the precipitation intensity, duration and the period between precipitation events, I would be curious to know if the rainfall patterns in Guizhou are similar to those in the dry season at Los Amigos, are the two sites comparable.

Line 161. Table S5, I haven't seen Tables S2, S3, and S4 referenced in the text.

Line 161. I think Martin Jiskra's 'A vegetation control on seasonal variations in global atmospheric mercury concentrations' merits a mention at this point.

Line 214. Could the authors briefly mention the fact that they Leaf Area Index measured here. I found myself reading about LIDAR measurements at line 692, what, why? and then discovering that the LAI was measured on line 816.

Line 278. The risk landscape it isn't immediately obvious if the second 'landscape' in this sentence refers to risk or geomorphology.

Line 284-8. Are the observed rates based on the back-on-the-envelope calculation (line 156)? If so, it would probably be appropriate to mention it again, I don't think it any way changes the conclusion that a significant amount of Hg is entering the regional and global atmospheric pools.

Line 673. All three seasons.. all three seasonal campaigns? The measurement periods aren't mentioned at all in the main body of the manuscript, is this an oversight? The length of the measurement periods should certainly be mentioned I think.

Line 687. This is the other instance of a number rather than a name based on 57, rather than Schulenberg (57).

Reviewer #2 (Remarks to the Author):

Summary

Many thanks for an interesting piece. The paper is sound scientifically, and I appreciate that this is probably the first time that a study has shown that mercury is intercepted by a canopy near artisanal gold mine sites. But I am really struggling to see the novelty here. I feel, in reading this manuscript, I have been thrown into a bit of a time machine, which I believe my comments below will offer some explanation as to why. Perhaps it is my bias and how weary I have grown with these scientific analyses which demonize artisanal gold mining, offering very little, beyond a glimpse of mercury and/or methylmercury loads in the environment. Maybe I am being a bit too harsh here but given where we are with managing mercury, and the readership and reach this journal has, I think this article needs to do much more than report contamination levels from ASGM activity in the Peruvian Amazon to warrant consideration for publication. My comments are below.

1. Whilst I understand that the objective here is to examine how ASGM activity is contributed to mercury and methylmercury contamination in the Peruvian Amazon, the authors provide little, if any, reference to what is being done to tackle it. For example, Peru ratified the Minamata Convention on Mercury in 2016. Did its initial mercury impact assessment capture the phenomena (canopy contamination) which the authors draw attention to in this paper? If not, then this is a significant policy oversight that needs to be mentioned.

2. ASGM is a significant livelihood activity in Peru, which, in the past decade in particular, has expanded into remote areas of the Amazon. There is a debate in the literature on this subject which the authors strangely ignore. As this is Nature Communications, it seems logical to me at least to frame this study by drawing upon this debate. Sure, it is an environmental study but readers are left hanging, struggling to understand why this problem persists. If informed about the importance of ASGM economically to millions in Peru, the phenomena examined take on a bit more meaning (a lot, in fact).

3. I am struggling to see the contribution to knowledge here. Is it new knowledge about the canopy in a region of the world where emissions/contamination levels of mercury emanating from ASGM have been analyzed for four decades? If so, without systematic engagement with the social context and debates on Minamata, I fail to see the value. If we compare what is being reported here to what was reported in the 1980s and 1990s, when people like Olaf Malm and Wolfgang Pfeiffer were reporting on mercury pollution attributable to gold rushes in the Amazon (albeit the Brazilian Amazon), there is not much new to sink our teeth into. In line with the objectives of the journal, are we, as readers, supposed to take away from this discussion that ASGM is inherently bad?

4. There is much more to say but what are the policy implications of this work? If we are to examine patterns of mercury contamination from ASGM reported from the region over the past four decades, are there any new developments we should know about that this work has

uncovered? We know that the Peruvian Amazon is a hotspot for all types of extractive industry activity, so where does ASGM fall on the list of development priorities in a country where the government seems to be receptive to developing pristine sections of the basin? What about the study itself: does it offer anything new in terms of methodology and approach to the body of scholarship on mercury contamination in ASGM? In other words, does this study offer us something new in terms of methodology and scientific approach that will help move us beyond the world of mundane hair sampling, blood sampling and urine analysis that remains the staples of this field after four decades?

5. As a study, this piece is fine and would be cited as is. But without reflecting on the wider policy implications of the work, the broader socioeconomic context of ASGM in the Peruvian Amazon and Minamata, the piece's home is in a scientific journal like *Environment International* or *The Science of the Total Environment*. But even these publications would demand some contextual background and recommendations.

Reviewer #3 (Remarks to the Author):

What are the noteworthy results?

This manuscript reports that mercury from 3 clandestine artisanal gold mining sites is accumulating in surrounding forest soils and probably wildlife due mainly to capture of vapors and dust by forest tree leaves followed by litter fall. A small sample of bird feathers suggests that soil accumulation is leading to food chain bioaccumulation. Sites that are deforested by mining operations actually accumulate less due to absence of canopy capture.

Will the work be of significance to the field and related fields?

Yes, although nothing in this manuscript is at all surprising, it is the first demonstration of the relationships in the Amazon, where this burgeoning illegal mining is on a collision course with the region's biodiversity.

How does it compare to the established literature? If the work is not original, please provide relevant references.

None of the hypotheses, that mining areas would have more mercury accumulation than non-mining area, that distance from mining area would predict accumulation, that tree canopy leaf area would be related to increased mercury deposition, and that birds in the area would accumulate mercury are original, but testing them in the Peruvian Amazon at actual mining sites is novel.

Does the work support the conclusions and claims, or is additional evidence needed?

Mostly, but the study is correlational in nature and some of the conclusions assert causation. For example:

Line 99: "affects" should be "correlates with"

Line 173: "due to" should be qualified with "presumably"

Line 212: "drives the pattern" needs to be qualified as it is earlier in the paragraph ("likely results from").

This sentence starting on Line 279 is confusing and unsupported: "The risk landscape of Hg from ASGM is therefore a function of both loading and landscape structure, suggesting differential impacts for the global Hg pool and terrestrial wildlife depending upon forest cover near mining." I believe they are saying that because loading and forest scrubbing both affect deposition that the amount of forest cover near mining will affect global deposition and local wildlife impacts. However, that is more than this paper should conclude because they have not examined wider deposition here, and wildlife impacts are not examined either (only a small study of bioaccumulation). They also point out that deforested areas can get lots of other input from spills, etc., so saying that global effects of a mine are determined by tree canopy, or that local wildlife impacts are, is unfounded and somewhat illogical. Keep conclusions to what was actually found - Hg is getting scrubbed down by the canopy and probably getting into local birds.

Are there any flaws in the data analysis, interpretation and conclusions? Do these prohibit publication or require revision?

Still publishable, but the tendency for overblown conclusions noted above is a serious flaw.

The other flaw, likely unfixable, is that there were only two reference sites and three mining sites,

of which only one of the mining sites was primary forest. Thus, authors must be careful to acknowledge the limits of their ability to generalize to other watersheds, the Amazon as a whole, or the globe. They generally are careful, restricting their conclusions to the whole of the Peruvian Amazon, but even that is not valid given the limited scope of the sampling sites, the correlational nature of the data, and the necessarily limited information on clandestine inputs.

Is the methodology sound? Does the work meet the expected standards in your field?
Yes, all QAQC, while not described in enough detail, indicates quality lab work.

Is there enough detail provided in the methods for the work to be reproduced?
Yes except that for bird study the text says "took advantage of study that was occurring" and then in Methods it is written up as if it were part of this study. The result is that it is never stated whether the bird sampling occurred during the same years as the rest of the study. Is it being published elsewhere, is that why they are vague, or was it added on later? A little more explanation would be helpful.

Title: Appropriate

Abstract: Does a good job of describing results but is vague about the bird study, for reasons I don't understand, as if it were a separate study. Also refers to "intact forest canopies" when in fact just one intact forest canopy was studied.

Introduction: The third hypothesis does not closely track the bird study: "(3) Is ASGM-derived Hg bioavailable to the terrestrial food web?" implies the whole food web was studied, and that it was studied in terms of bio-availability. In fact only a few bird feathers (<20) were sampled and none of their prey or any other component of the food web and so it should be rephrased to match what question was actually asked: "Is there any evidence that bioaccumulation of mercury is occurring in forest-dwelling resident birds?"

Results and Discussion:

Line 155: "back of the envelope calculations" is a vague term - too vague for technical writing like this, and begs the question of why not regular calculations?

Methods: Well done.

Figures: These are nicely done, necessary and helpful.

REVIEWER COMMENTS

Reviewer #1 (Remarks to the Author):

This article addresses an issue of major concern within the field on environmental mercury studies. Unfortunately it seems that ASGM is increasing and unlikely to diminish in the near future. The fact that according to the most recent emission inventories it has overtaken power production from fossil fuels, as the authors point out, has effectively shifted the major emission source of atmospheric mercury from the Global North to the Global South, which the authors also point out, but may want to emphasize a little more in the Abstract. This shift also marks a move from the major mercury emission sources being located at mid-latitudes to more tropical regions, with consequences for Hg dispersion and deposition patterns.

This manuscript documents detailed measurements of mercury in a tropical ecosystem close to ASGM activities, it is a significant contribution to the field of environmental mercury cycling, and addresses a pressing problem in the field.

The article is well written and clearly presented, and I have only a few pedantic and stylistic points to make, I thoroughly recommend publication with very minor revision.

In the paragraph beginning on line 78, regarding the burning off of Hg from amalgam. This is often done without a retort to recover Hg, however efforts from the UNEP global mercury partnership, UNIDO and a number of NGOs are making efforts to encourage miners to improve practices, and I think it would be good to mention these efforts somehow.

We have added a detailed description of these efforts. The text now reads:

Page 4, Lines 86-102: “The international Minamata Convention on Mercury entered into force in 2017 with Article 7 specifically directed at Hg releases from ASGM. In ASGM, liquid elemental Hg is added to sediment or ores to isolate gold. This amalgam is subsequently heated, which concentrates the gold and releases gaseous elemental Hg (GEM; Hg⁰) into the atmosphere. Amalgam burning often occurs without a retort or other Hg-capturing device, though efforts are being undertaken by groups such as the United Nations Environment Program (UNEP) Global Mercury Partnership, United Nations Industrial Development Organization (UNIDO), and non-governmental organizations to encourage miners to mitigate Hg emissions. As of this writing in 2021, 132 countries including Peru, have signed onto the Minamata Convention. While each party is required to create a National Action Plan and to specifically address Hg reductions in association with ASGM, these plans are still in development. These plans include Hg risks associated with ASGM near aquatic ecosystems, involving miners and those living near amalgam burning, and involving communities that consume large quantities predatory fish. Occupational Hg exposure via inhalation of Hg vapors from amalgam burning, dietary Hg exposure via the consumption of fish, and bioaccumulation of Hg in the aquatic food web have been the focus of most scientific studies related to ASGM, including early studies in the Amazon (e.g., see Lodenius and Malm¹⁵).”

Line 87. I don't know if it's the journal's policy, but 'see summary table in 16' is horrible. Is it not

possible to write 'see summary table in Gersoin et al., (16)'? This occurs in a few other places in the manuscript.

We thank the reviewer for pointing this out. We have made the suggested wording change here and in other instances where it occurs in the manuscript.

Line 108. It may not be clear to readers why throughfall, which occurs when precipitation effectively washes previously deposited Hg from the canopy and deposits it to the forest floor is considered to be dry deposition, it may be an idea to cite the Wright paper here (ref 28) and add a short comment summarising the description in the reference "Mercury in throughfall includes wet-deposited Hg above the canopy and a portion of dry-deposited Hg washed off from the canopy. Thus, litterfall Hg likely represents the low end of Hg dry deposition when Hg emission from the underlying soil is small, although it can be higher than the actual dry deposition above the canopy due to the interception of emitted Hg by forest leaves if soil Hg emissions are high and the ambient Hg concentrations above the forests are low. The difference between throughfall and open-area wet deposition, with the former collected inside and the latter outside a forest, should also represent a portion of the dry deposition. Thus, on annual basis, dry deposition can be approximated as the sum of litterfall and throughfall Hg minus wet-deposited Hg." This is mentioned later, line 188, but I think it would be useful to clarify the concept with the reader at the start.

We have added in two sentences to clarify how wet and dry deposition are determined. The text reads:

Page 5, Lines 109-111: "Wet deposition can be determined by Hg flux in precipitation collected in clearings. Dry deposition can be determined by the sum of Hg flux in litterfall and throughfall minus Hg flux in precipitation."¹⁷

Line 116. Not everybody knows what placer deposits are, a short description would avoid the reader skipping to wikipedia whilst reading (the same probably applies to neotropical, but it's in the abstract so perhaps there's a word limit).

We have deleted the word "placer." We have also changed "neotropical" to "tropical" in the abstract.

Line 122. Are the 'remote' sites upwind of where ASGM activity is occurring, is that why they can be considered remote? 50km downwind would seem relatively close to me.

Unfortunately, there are no available data at an adequate spatial resolution to reliably determine the wind direction at these sites. The remote sites do not have any ASGM activity occurring and are also 50-100 km from population centers.

A large part of this first paragraph of the Results and Discussion is repeated pretty much word for word in the Methods (Sample Collection) section. Probably one could make reference to the other.

While we agree that much of this first paragraph of the Results and Discussion is included in the Methods section, we have elected to retain most of it within the manuscript. We have deleted information from the paragraph in the Results and Discussion that are not crucial to interpreting our results. The information included now is vital to understanding the results presented. Since the Methods section is long and per Nature Communications style is found at the end of the manuscript, we want to make sure that this information is clear to the reader.

Line 151. The concentration of total Hg in throughfall will be in part determined by the precipitation intensity, duration and the period between precipitation events, I would be curious to know if the rainfall patterns in Guizhou are similar to those in the dry season at Los Amigos, are the two sites comparable.

Both Los Amigos and Guizhou have humid, monsoonal climates. Guizhou, China receives ~1080 mm annual precipitation, compared to ~2500 mm at Los Amigos. Patterns of precipitation are similar in both locations. At both sites, 3-4 months of the year account for about half of the annual precipitation. Thus, despite receiving double the amount of precipitation, the concentration of Hg in throughfall at Los Amigos is on par with that in Guizhou. We have modified this sentence to address this:

Page 8, Lines 182-186: “The average concentration of total Hg in throughfall at Los Amigos Conservation Concession during the dry season was among the highest reported in the literature (range: 18-61 ng L⁻¹), rivaling levels measured in sites contaminated from cinnabar mining and industrial coal combustion in Guizhou, China when considering differences in precipitation volume.”²⁴”

Line 161. Table S5, I haven't seen Tables S2, S3, and S4 referenced in the text.

We don't reference tables S2-S4 in the text. Table S5 is placed after Tables S2-S4 since Table S5 (and subsequent supplementary Tables) are very long tables with chemistry information.

Line 161. I think Martin Jiskra's 'A vegetation control on seasonal variations in global atmospheric mercury concentrations' merits a mention at this point.

We have added in the suggested reference. The text now reads:

Page 8, Lines 190-194: “Other than Hg, only aluminum and manganese were elevated in throughfall at the mining sites, which likely is due to land clearing associated with mining; all other measured major and trace elements did not vary between the mining and remote locations (Table S5), a finding consistent with leaf Hg dynamics²⁵ and ASGM amalgam burning, rather than airborne dust, as the main source of Hg in throughfall.”

Line 214. Could the authors briefly mention the fact that they Leaf Area Index measured here. I found myself reading about LIDAR measurements at line 692, what, why? and then discovering that the LAI was measured on line 816.

We have reworded this sentence to mention leaf area index. The text now reads:

Page 11, Lines 247-249: “Since Hg inputs from ASGM in the Peruvian Amazon are largely deposited to terrestrial ecosystems via interaction with the forest canopy, we tested whether higher canopy density (i.e., leaf area index) would lead to higher Hg inputs.”

We have also clarified the use of LIDAR measurements so that LAI measurements are apparent earlier in the text. The text now reads:

Page 21, Lines 483-486: “To determine leaf area index (LAI), lidar data were collected using the GatorEye Uninhabited Flying Laboratory, which is a sensor fusion drone system (details available at www.gatoreye.org, with plot scale data download also available using the “2019 Peru Los Amigos June” link).⁶²”

Line 278. The risk landscape it isn't immediately obvious if the second 'landscape' in this sentence refers to risk or geomorphology.

*We have reworded this sentence to make the meaning clearer. The sentence now reads:
Page 14, Lines 319-323: "The risk to the landscape of Hg from ASGM is therefore a function not only of direct Hg inputs via atmospheric emissions, spills, and tailings, but also of the landscape's potential to capture, store, and transform Hg into the more bioavailable form of MeHg, suggesting differential impacts for the global Hg pool and terrestrial wildlife depending upon forest cover near mining."*

Line 284-8. Are the observed rates based on the back-on-the-envelope calculation (line 156)? If so, it would probably be appropriate to mention it again, I don't think it any way changes the conclusion that a significant amount of Hg is entering the regional and global atmospheric pools. *These observed rates are based on the same calculation. However, per Reviewer 3's comments, we have removed the phrase "back-of-the-envelope calculation" from the manuscript.*

Line 673. All three seasons.. all three seasonal campaigns? The measurement periods aren't mentioned at all in the main body of the manuscript, is this an oversight? The length of the measurement periods should certainly be mentioned I think.

We have reworded "all three seasons" to "all three seasonal campaigns" in the three instances that we use this phrase in the Methods. We have also added the measurement periods to the main body of the manuscript. It now reads:

Page 7, Lines 160-164: "At each location, we installed deposition samplers both in clearings (deforested areas completely void of woody plants) and beneath the tree canopy (forested areas) in the dry and wet seasons for a total of three seasonal campaigns (each 1-2 months in duration) to collect wet deposition and throughfall, respectively, as well as passive air samplers in clearings to collect GEM."

Line 687. This is the other instance of a number rather than a name based on 57, rather than Schulenberg (57).

We have made this suggested change to the text.

Reviewer #2 (Remarks to the Author):

Summary

Many thanks for an interesting piece. The paper is sound scientifically, and I appreciate that this is probably the first time that a study has shown that mercury is intercepted by a canopy near artisanal gold mine sites. But I am really struggling to see the novelty here. I feel, in reading this manuscript, I have been thrown into a bit of a time machine, which I believe my comments below will offer some explanation as to why. Perhaps it is my bias and how weary I have grown with these scientific analyses which demonize artisanal gold mining, offering very little, beyond a glimpse of mercury and/or methylmercury loads in the environment. Maybe I am being a bit too harsh here but given where we are with managing mercury, and the readership and reach this journal has, I think this article needs to do much more than report contamination levels from ASGM activity in the Peruvian Amazon to warrant consideration for publication. My comments are below.

1. Whilst I understand that the objective here is to examine how ASGM activity is contributed to mercury and methylmercury contamination in the Peruvian Amazon, the authors provide little, if any, reference to what is being done to tackle it. For example, Peru ratified the Minamata Convention on Mercury in 2016. Did its initial mercury impact assessment capture the phenomena (canopy contamination) which the authors draw attention to in this paper? If not, then this is a significant policy oversight that needs to be mentioned.

We thank the reviewer for this suggestion and agree that it is important to mention international measures targeted at reducing Hg from ASGM. While countries are assessing Hg impact, canopy transfer and terrestrial contamination have largely been ignored. Most emphasis has been placed on aquatic ecosystems. We have clarified this in the text:

Page 4, Lines 86-102: “The international Minamata Convention on Mercury entered into force in 2017 with Article 7 specifically directed at Hg releases from ASGM. In ASGM, liquid elemental Hg is added to sediment or ores to isolate gold. This amalgam is subsequently heated, which concentrates the gold and releases gaseous elemental Hg (GEM; Hg⁰) into the atmosphere. Amalgam burning often occurs without a retort or other Hg-capturing device, though efforts are being undertaken by groups such as the United Nations Environment Program (UNEP) Global Mercury Partnership, United Nations Industrial Development Organization (UNIDO), and non-governmental organizations to encourage miners to mitigate Hg emissions. As of this writing in 2021, 132 countries including Peru, have signed onto the Minamata Convention. While each party is required to create a National Action Plan and to specifically address Hg reductions in association with ASGM, these plans are still in development. These plans include Hg risks associated with ASGM near aquatic ecosystems, involving miners and those living near amalgam burning, and involving communities that consume large quantities of predatory fish. Occupational Hg exposure via inhalation of Hg vapors from amalgam burning, dietary Hg exposure via the consumption of fish, and bioaccumulation of Hg in the aquatic food web have been the focus of most scientific studies related to ASGM, including early studies in the Amazon (e.g., see Lodenius and Malm¹⁵).”

2. ASGM is a significant livelihood activity in Peru, which, in the past decade in particular, has expanded into remote areas of the Amazon. There is a debate in the literature on this subject

which the authors strangely ignore. As this is Nature Communications, it seems logical to me at least to frame this study by drawing upon this debate. Sure, it is an environmental study but readers are left hanging, struggling to understand why this problem persists. If informed about the importance of ASGM economically to millions in Peru, the phenomena examined take on a bit more meaning (a lot, in fact).

We agree that ASGM is a main source of income for many miners in this region. We have added information to the Introduction that states the importance of ASGM as a livelihood. We have also added information to the Conclusion that states the importance of socioeconomic investments to reduce the financial pressures that lead people to mine. We believe that this provides better context to the reader, as suggested by the reviewer.

In the Introduction, we have added:

Page 3, Lines 74-77: “While ASGM is an important livelihood for local communities, it results in widespread deforestation,^{2,3} extensive conversion of forests to ponds,⁴ high sediment loading in nearby rivers,^{5,6} and is the largest global source of atmospheric mercury (Hg) emissions and freshwater releases.⁷”

In the Conclusion, we have added:

Page 17, Lines 385-387: “Those efforts include a variety of approaches that range from the relatively simple Hg-capture system of retorts to the more challenging economic and social investments that would reduce the financial incentives of this illegal activity as a livelihood.”

3. I am struggling to see the contribution to knowledge here. Is it new knowledge about the canopy in a region of the world where emissions/contamination levels of mercury emanating from ASGM have been analyzed for four decades? If so, without systematic engagement with the social context and debates on Minamata, I fail to see the value. If we compare what is being reported here to what was reported in the 1980s and 1990s, when people like Olaf Malm and Wolfgang Pfeiffer were reporting on mercury pollution attributable to gold rushes in the Amazon (albeit the Brazilian Amazon), there is not much new to sink our teeth into. In line with the objectives of the journal, are we, as readers, supposed to take away from this discussion that ASGM is inherently bad?

The work conducted by Olaf Malm and Wolfgang Pfeiffer was foundational in establishing Hg contamination from mining as an important source of Hg exposure to people. However, their work as well as other work conducted in the Amazon near ASGM has focused on direct Hg exposure to miners through inhalation of Hg vapors and exposure via consumption of fish. This study is novel in that it includes the first measurements of methyl Hg in terrestrial ecosystems in the region and is the first to document Hg concentrations in deposition near ASGM globally. Our data show that Hg is being captured by intact forests, being methylated into the toxic and bioavailable form of MeHg, and entering into terrestrial food webs. While these results would be expected, we are the first to show that concerns for Hg exposure from ASGM extend not just to the aquatic ecosystem but also to the terrestrial ecosystem, and we are the first to quantify the fluxes of the different Hg deposition pathways to the terrestrial landscape near ASGM. While this forest scrubbing effect has been observed in boreal and temperate forests, no study has yet measured this effect in tropical forests near such a large Hg emission source. We also show that the magnitude of the throughfall and litterfall flux is surprisingly much larger than the

precipitation flux. The observed seasonality in Hg deposition is also a novel contribution. We have added this information to the text.

In the Introduction, we have added:

Page 4, Lines 99-102: “Occupational Hg exposure via inhalation of Hg vapors from amalgam burning, dietary Hg exposure via the consumption of fish, and bioaccumulation of Hg in the aquatic food web have been the focus of most scientific studies related to ASGM, including early studies in the Amazon (e.g., see Lodenius and Malm¹⁵).”

Page 5, Lines 119-127: “Since two of the three pathways by which Hg is deposited onto the landscape (throughfall and litterfall) are dependent upon Hg interaction with plant surfaces, we would also anticipate the rate of Hg deposition into ecosystems and the risk it poses to animals to be heavily influenced by the structure of vegetation, as suggested by observations in boreal and temperate forests at northern latitudes.¹⁹ However, we also recognize that ASGM activities frequently occur in tropical landscapes, where the canopy structure and relative abundance of exposed leaf area are vastly different. The relative importance of Hg deposition pathways in these ecosystems has yet to be firmly quantified, particularly for forests in proximity to Hg emission sources with an intensity rarely observed in northern forests.”

In the Results and Discussion, we have added:

Page 10, Lines 219-221: “Taken together, approximately 94% of total Hg deposited in conserved forests at Los Amigos occurs via dry deposition (throughfall + litterfall – precipitation Hg), a much higher contribution from dry deposition than most other forested landscapes globally.”

In the Conclusion, we have added:

Page 13, Lines 301-305: “The most important and novel implication of our work is the documentation of elevated quantities of Hg being delivered to forests near ASGM activity. Our data show that this Hg is available to, and moving through, terrestrial food webs. Moreover, very large quantities of Hg are stored in biomass and soils with the potential for release with land use change⁴ and forest fires.^{41,42}”

4. There is much more to say but what are the policy implications of this work? If we are to examine patterns of mercury contamination from ASGM reported from the region over the past four decades, are there any new developments we should know about that this work has uncovered? We know that the Peruvian Amazon is a hotspot for all types of extractive industry activity, so where does ASGM fall on the list of development priorities in a country where the government seems to be receptive to developing pristine sections of the basin? What about the study itself: does it offer anything new in terms of methodology and approach to the body of scholarship on mercury contamination in ASGM? In other words, does this study offer us something new in terms of methodology and scientific approach that will help move us beyond the world of mundane hair sampling, blood sampling and urine analysis that remains the staples of this field after four decades?

Please see our response to comment 3 regarding the novelty of our results. We have added additional information regarding the implications of our findings to the final paragraph. It reads:

Page 16, Lines 372-374: “These results highlight the importance of preventing ASGM activity from occurring within national reserves and the buffer zones that surround them.”

Page 17, Lines 383-387: “Our finding that terrestrial biota may be at considerable risk from ASGM-derived Hg pollution should provide further incentive for on-going efforts to reduce the release of Hg from ASGM. Those efforts include a variety of approaches that range from the relatively simple Hg-capture system of retorts to the more challenging economic and social investments that would reduce the financial incentives of this illegal activity as a livelihood.”

5. As a study, this piece is fine and would be cited as is. But without reflecting on the wider policy implications of the work, the broader socioeconomic context of ASGM in the Peruvian Amazon and Minamata, the piece’s home is in a scientific journal like Environment International or The Science of the Total Environment. But even these publications would demand some contextual background and recommendations.

Please see the changes we have made to the manuscript to address comments 1-4. We believe these edits incorporate policy implications of the work and the broader socioeconomic context of ASGM and enhance the manuscript overall.

Reviewer #3 (Remarks to the Author):

What are the noteworthy results?

This manuscript reports that mercury from 3 clandestine artisanal gold mining sites is accumulating in surrounding forest soils and probably wildlife due mainly to capture of vapors and dust by forest tree leaves followed by litter fall. A small sample of bird feathers suggests that soil accumulation is leading to food chain bioaccumulation. Sites that are deforested by mining operations actually accumulate less due to absence of canopy capture.

Will the work be of significance to the field and related fields?

Yes, although nothing in this manuscript is at all surprising, it is the first demonstration of the relationships in the Amazon, where this burgeoning illegal mining is on a collision course with the region's biodiversity.

We found that some of our results were, in fact, surprising. The magnitude of Hg deposition to intact Amazonian forests was surprisingly high, with throughfall concentrations that are on par with cities in China engaged in both cinnabar mining and industrial coal combustion, and annual throughfall fluxes that are the highest ever measured. We were also surprised by the very large magnitude of litterfall and throughfall relative to precipitation flux. Finally, we were surprised by the amount of MeHg observed in upland soils, which suggests in situ Hg methylation in these soils.

In the text, we have stated:

Page 8, Lines 182-189: “The average concentration of total Hg in throughfall at Los Amigos Conservation Concession during the dry season was among the highest reported in the literature (range: 18-61 ng L⁻¹), rivaling levels measured in sites contaminated from cinnabar mining and industrial coal combustion in Guizhou, China when considering differences in precipitation volume.²⁴ These values represent, to our knowledge, the largest measured annual throughfall Hg flux, according to calculations using Hg concentrations and precipitation rates from both the dry and wet seasons (71 μg m⁻² yr⁻¹; Table S1).”

Page 10, Lines 219-221: “Taken together, approximately 94% of total Hg deposited in conserved forests at Los Amigos occurs via dry deposition (throughfall + litterfall – precipitation Hg), a much higher contribution from dry deposition than most other forested landscapes globally.”

Pages 12-13, Lines 283-287: “However, we document for the first time that there are measurable MeHg concentrations within Amazonian soils near ASGM, suggesting that elevated MeHg concentrations extend beyond aquatic ecosystems and into terrestrial environments within these ASGM-impacted areas, including soils that are inundated during the wet season as well as those that remain dry throughout the year.”

How does it compare to the established literature? If the work is not original, please provide relevant references.

None of the hypotheses, that mining areas would have more mercury accumulation than non-mining area, that distance from mining area would predict accumulation, that tree canopy leaf area would be related to increased mercury deposition, and that birds in the area would

accumulate mercury are original, but testing them in the Peruvian Amazon at actual mining sites is novel.

Does the work support the conclusions and claims, or is additional evidence needed?

Mostly, but the study is correlational in nature and some of the conclusions assert causation. For example:

Line 99: "affects" should be "correlates with"

We have made this suggested change.

Line 173: "due to" should be qualified with "presumably"

We have made this suggested change.

Line 212: "drives the pattern" needs to be qualified as it is earlier in the paragraph ("likely results from").

We have changed "drives the patterns" to "likely leads to patterns," so that the sentence now reads:

Page 11, Lines 244-246: "This production of Hg and dust in the dry season likely leads to patterns in Hg flux within deforested compared to forested areas at the Los Amigos Conservation Concession."

This sentence starting on Line 279 is confusing and unsupported: "The risk to the landscape of Hg from ASGM is therefore a function of both loading and landscape structure, suggesting differential impacts for the global Hg pool and terrestrial wildlife depending upon forest cover near mining." I believe they are saying that because loading and forest scrubbing both affect deposition that the amount of forest cover near mining will affect global deposition and local wildlife impacts. However, that is more than this paper should conclude because they have not examined wider deposition here, and wildlife impacts are not examined either (only a small study of bioaccumulation). They also point out that deforested areas can get lots of other input from spills, etc., so saying that global effects of a mine are determined by tree canopy, or that local wildlife impacts are, is unfounded and somewhat illogical. Keep conclusions to what was actually found - Hg is getting scrubbed down by the canopy and probably getting into local birds.

We are suggesting that both loading and forest scrubbing of atmospheric Hg affect Hg exposure to wildlife. Our results show that intact forested canopies near ASGM both have higher atmospheric Hg concentrations and higher ability to capture that Hg (via dry deposition and uptake into leaves). We have reworded this sentence so that we are not exaggerating our conclusions. It now reads:

Page 14, Lines 319-323: "The risk to the landscape of Hg from ASGM is therefore a function not only of direct Hg inputs via atmospheric emissions, spills, and tailings, but also of the landscape's potential to capture, store, and transform Hg into the more bioavailable form of MeHg, suggesting differential impacts for the global Hg pool and terrestrial wildlife depending upon forest cover near mining."

Are there any flaws in the data analysis, interpretation and conclusions? Do these prohibit publication or require revision?

Still publishable, but the tendency for overblown conclusions noted above is a serious flaw.

We have reviewed the manuscript to ensure that we use verbs appropriate to the correlational nature of our study. We have also made the suggested wording changes of the reviewer, listed above.

The other flaw, likely unfixable, is that there were only two reference sites and three mining sites, of which only one of the mining sites was primary forest. Thus, authors must be careful to acknowledge the limits of their ability to generalize to other watersheds, the Amazon as a whole, or the globe. They generally are careful, restricting their conclusions to the whole of the Peruvian Amazon, but even that is not valid given the limited scope of the sampling sites, the correlational nature of the data, and the necessarily limited information on clandestine inputs.

We have made an effort throughout the manuscript to be as transparent as possible about the data from which our inferences are derived. We have edited the language throughout the text to ensure it is clear that we are referring to only one intact forested site near ASGM. We have also made all of the suggested wording changes of this reviewer, as detailed above. While we agree that more data and more sites would be useful, we do not think it would change the conclusions drawn here.

Is the methodology sound? Does the work meet the expected standards in your field?

Yes, all QAQC, while not described in enough detail, indicates quality lab work.

Based on the comment below regarding our Methods and since there is no information about what additional detail the reviewer would like to see, we haven't made any changes to the methodology. For gaseous elemental Hg, we performed continuous calibration verification and quality control standards, with all values falling within 5% of accepted values. For total Hg in water, we performed continuous calibration verification, with all values falling within 15% of accepted values. For total Hg in solids, we performed continuous calibration verification, quality control standards, and matrix spikes, with all values falling within 10% of accepted values. We also ran all samples in duplicate and only accepted duplicates with <15% relative percent difference. For MeHg, we performed continuous calibration verification and quality control standards, with all values falling within 15% of accepted values. For ancillary chemical analyses, all standards were within 10% of accepted values. Detailed information on QAQC can be found in the Methods section.

Is there enough detail provided in the methods for the work to be reproduced?

Yes except that for bird study the text says "took advantage of study that was occurring" and then in Methods it is written up as if it were part of this study. The result is that it is never stated whether the bird sampling occurred during the same years as the rest of the study. Is it being published elsewhere, is that why they are vague, or was it added on later? A little more explanation would be helpful.

Co-authors on this manuscript led the bird sampling, which examined other bird traits beyond just Hg concentrations. The bird sampling that occurred took place during the same time interval as the rest of the study. We have deleted the text that stated "took opportunistic advantage of two bird surveys," so that this information is clearer.

Title: Appropriate

Abstract: Does a good job of describing results but is vague about the bird study, for reasons I

don't understand, as if it were a separate study. Also refers to "intact forest canopies" when in fact just one intact forest canopy was studied.

We have changed "intact forest canopies" to "intact forest canopy." We also have clarified the bird study in the Results and Discussion section to show that the bird study is part of this study. The text now reads:

Page 16, Lines 356-360: "To assess whether the Hg deposited into these forested areas are entering terrestrial food webs, we measured total Hg concentrations in the tail feathers of several species of resident songbirds from the Los Amigos Conservation Concession (mining-impacted) and Cocha Cashu Biological Station (unimpacted old growth forest), a remote site 140 km beyond our most upstream sampling site at Boca Manu."

Introduction: The third hypothesis does not closely track the bird study: "(3) Is ASGM-derived Hg bioavailable to the terrestrial food web?" implies the whole food web was studied, and that it was studied in terms of bio-availability. In fact only a few bird feathers (<20) were sampled and none of their prey or any other component of the food web and so it should be rephrased to match what question was actually asked: "Is there any evidence that bioaccumulation of mercury is occurring in forest-dwelling resident birds?"

We agree that this hypothesis was too broad. We have reworded it to:

Page 6, Lines 130-131: "Is there evidence that Hg bioaccumulation is elevated in forest-dwelling resident songbirds near ASGM activity?"

Results and Discussion:

Line 155: "back of the envelope calculations" is a vague term - too vague for technical writing like this, and begs the question of why not regular calculations?

We have deleted the phrase "back-of-the-envelope calculation" from the manuscript. All values presented in this manuscript use the Hg concentrations obtained in our measurements.

Methods: Well done.

Figures: These are nicely done, necessary and helpful.

REVIEWERS' COMMENTS

Reviewer #1 (Remarks to the Author):

The authors have replied to all my questions and suggestions adequately. They have revised the paper accordingly, and I think the article may now be published.

Reviewer #2 (Remarks to the Author):

Dear author,

Many thanks for the revision. You have, indeed, addressed the comments but I really worry about the message being communicated. Specifically, as this manuscript is being considered for Nature Communications, my concern is just this.

I have asked the authors to be more specific with clarifying their contribution to knowledge and they have. But in doing so, and using words such as 'threat' instead of more neutral words like 'challenge', they run the risk of portraying a sector in a very negative way. It really sends the wrong message and is really perplexing for people like myself who have been trying to project a more 'balanced' view of small-scale mining for years. How, then, do we fix this? I think the best way is to tone down the language and convey that ASM poses some environmental concerns in Peru and elsewhere but in recognition of its contribution to livelihoods, Minamata offers us a vehicle to take into consideration, seriously, studies such as this one on the canopy whilst taking action which also considers the importance of the sector to people's lives.

So it brings me back to Minamata, which has been covered here in the revised but not particularly comprehensively. So in addition to toning down the language, and not leaving the layreader with the impression that we should eradicate all small-scale mining activity worldwide perhaps the authors could engage a bit more with the aims of the Convention, particularly the NAP, which seeks to retain focus on the human and social side of ASM whilst tackling its mercury problem. Have a look at, and cross-reference some of the studies which illuminate these elements. See the following:

Hilson, G., Hu, Y., Kumah, C. Locating female 'Voices' in the Minamata Convention on Mercury in Sub-Saharan Africa: The case of Ghana (2020) *Environmental Science and Policy*, 107, pp. 123-136.

Stylo, M., De Haan, J., Davis, K. Collecting, managing and translating data into National Action Plans for artisanal and small scale gold mining (2020) *Extractive Industries and Society*, 7 (1), pp. 237-248.

Hilson, G., Zolnikov, T.R., Ortiz, D.R., Kumah, C. Formalizing artisanal gold mining under the Minamata convention: Previewing the challenge in Sub-Saharan Africa (2018) *Environmental Science and Policy*, 85, pp. 123-131.

Spiegel, S., Keane, S., Metcalf, S., Veiga, M. Implications of the minamata convention on mercury for informal gold mining in sub-saharan africa: From global policy debates to grassroots implementation? (2015) *Environment, Development and Sustainability*, 17 (4), pp. 765-785.

Clifford, M.J. Future strategies for tackling mercury pollution in the artisanal gold mining sector: Making the Minamata Convention work (2014) *Futures*, 62, pp. 106-112.

I think if these changes are made, then the manuscript will be suitable for the journal.

Reviewer #3 (Remarks to the Author):

Edits were made in response to all of my relevant comments, and were satisfactory. I believe the same is true for both other reviewers, in my opinion, based on the rebuttal letter and a re-reading of the manuscript. I feel like the authors have done as much as they can to provide context and

describe contribution as novelty, and doing more would cross the line into policy or propaganda.

REVIEWERS' COMMENTS

Reviewer #1 (Remarks to the Author):

The authors have replied to all my questions and suggestions adequately. They have revised the paper accordingly, and I think the article may now be published.

We appreciate these remarks from the reviewer.

Reviewer #2 (Remarks to the Author):

Dear author,

Many thanks for the revision. You have, indeed, addressed the comments but I really worry about the message being communicated. Specifically, as this manuscript is being considered for Nature Communications, my concern is just this.

I have asked the authors to be more specific with clarifying their contribution to knowledge and they have. But in doing so, and using as words such as 'threat' instead of more neutral words like 'challenge', they run the risk of portraying a sector in a very negative way. It really sends the wrong message and is really perplexing for people like myself who have been trying to project a more 'balanced' view of small-scale mining for years. How, then, do we fix this? I think the best way is to tone down the language and convey that ASM poses some environmental concerns in Peru and elsewhere but in recognition of its contribution to livelihoods, Minamata offers us a vehicle to take into consideration, seriously, studies such as this one on the canopy whilst taking action which also considers the importance of the sector to people's lives.

So it brings me back to Minamata, which has been covered here in the revised but not particularly comprehensively. So in addition to toning down the language, and not leaving the layreader with the impression that we should eradicate all small-scale mining activity worldwide perhaps the authors could engage a bit more with the aims of the Convention, particularly the NAP, which seeks to retain focus on the human and social side of ASM whilst tackling its mercury problem. Have a look at, and cross-reference some of the studies which illuminate these elements. See the following:

Hilson, G., Hu, Y., Kumah, C. Locating female 'Voices' in the Minamata Convention on Mercury in Sub-Saharan Africa: The case of Ghana (2020) *Environmental Science and Policy*, 107, pp. 123-136.

Stylo, M., De Haan, J., Davis, K. Collecting, managing and translating data into National Action Plans for artisanal and small scale gold mining (2020) *Extractive Industries and Society*, 7 (1), pp. 237-248.

Hilson, G., Zolnikov, T.R., Ortiz, D.R., Kumah, C. Formalizing artisanal gold mining under the Minamata convention: Previewing the challenge in Sub-Saharan Africa (2018) *Environmental Science and Policy*, 85, pp. 123-131.

Spiegel, S., Keane, S., Metcalf, S., Veiga, M. Implications of the minamata convention on mercury for informal gold mining in sub-saharan africa: From global policy debates to grassroots

implementation? (2015) *Environment, Development and Sustainability*, 17 (4), pp. 765-785.

Clifford, M.J. Future strategies for tackling mercury pollution in the artisanal gold mining sector: Making the Minamata Convention work (2014) *Futures*, 62, pp. 106-112.

I think if these changes are made, then the manuscript will be suitable for the journal.

We appreciate these suggestions from the reviewer. We have toned down the language, changing “threat” to “challenge” and “illegal to informal.” We have also added in more context regarding the social and human dimensions of ASGM, as suggested by the reviewer, including some of the provided references. The changes we have implemented are:

Lines 72-73: “A growing challenge to tropical forested ecosystems is artisanal and small-scale gold mining (ASGM).”

Lines 93-100: “As of this writing in 2021, 132 countries including Peru, have signed onto the Minamata Convention and have begun to develop National Action Plans to specifically address Hg reductions in association with ASGM. Scholars have called on these National Action Plans to be inclusive, ongoing, and holistic, considering both socioeconomic drivers and environmental harms.¹⁵⁻¹⁸ Current plans to address the consequences of Hg in the environment focus on Hg risks associated with ASGM near aquatic ecosystems, involving miners and those living near amalgam burning, and involving communities that consume large quantities of predatory fish.”

Lines 157-161: “Note that the release of Hg vapor from the burning of Hg-gold amalgams regularly occurs within this mining zone, at both the Boca Colorado and Laberinto sites, though exact locations and number of locations are unknown since these activities are generally informal and clandestine; we refer to mining and amalgam burning collectively as “ASGM activity.””

Lines 357-358: “Formalizing ASGM activity^{15,16} could be a mechanism for ensuring protected lands are not mined.”

Lines 388-390: “Those efforts include a variety of approaches that range from the relatively simple Hg-capture system of retorts to the more challenging economic and social investments that would formalize this activity and reduce the financial incentives of conducting ASGM illegally.”

Reviewer #3 (Remarks to the Author):

Edits were made in response to all of my relevant comments, and were satisfactory. I believe the same is true for both other reviewers, in my opinion, based on the rebuttal letter and a re-reading of the manuscript. I feel like the authors have done as much as they can to provide context and describe contribution as novelty, and doing more would cross the line into policy or propaganda.

We appreciate these remarks from the reviewer.